# Naturally-Sourced Antibacterial Polymeric Nanomaterials with Special Reference to Modified Polymer Variants

**DOI:** 10.3390/ijms23084101

**Published:** 2022-04-07

**Authors:** Marian Rofeal, Fady Abdelmalek, Alexander Steinbüchel

**Affiliations:** 1International Center for Research on Innovative Biobased Materials (ICRI-BioM)—International Research Agenda, Lodz University of Technology, Zeromskiego 116, 90–924 Lodz, Poland; fady.abdelmalek@p.lodz.pl; 2Department of Botany and Microbiology, Faculty of Science, Alexandria University, Alexandria 21521, Egypt

**Keywords:** polymeric nanomaterials, drug delivery, antibacterial, post-synthetic modification, nanocarriers, multidrug resistance

## Abstract

Despite the recent advancements in treating bacterial infections, antibiotic resistance (AR) is still an emerging issue. However, polymeric nanocarriers have offered unconventional solutions owing to their capability of exposing more functional groups, high encapsulation efficiency (EE) and having sustained delivery. Natural polymeric nanomaterials (NMs) are contemplated one of the most powerful strategies in drug delivery (DD) in terms of their safety, biodegradability with almost no side effects. Every nanostructure is tailored to enhance the system functionality. For example, cost-effective copper NPs could be generated in situ in cellulose sheets, demonstrating powerful antibacterial prospects for food safety sector. Dendrimers also have the capacity for peptide encapsulation, protecting them from proteolytic digestion for prolonged half life span. On the other hand, the demerits of naturally sourced polymers still stand against their capacities in DD. Hence, Post-synthetic modification of natural polymers could play a provital role in yielding new hybrids while retaining their biodegradability, which could be suitable for building novel super structures for DD platforms. This is the first review presenting the contribution of natural polymers in the fabrication of eight polymeric NMs including particulate nanodelivery and nanofabrics with antibacterial and antibiofilm prospects, referring to modified polymer derivatives to explore their full potential for obtaining sustainable DD products.

## 1. Introduction

In the recent decades, owing to the rise of antibiotic-resistant organisms, traditional antibiotic therapies are becoming more ineffective in combating bacterial infections. It’s also worth noting that the development of bacterial resistance outpaces the discovery of new antibiotics [1]. On the other hand, nanotechnology has opened up new avenues for tackling problems in a variety of industries. Numerous polymeric-based NMs including polymeric nanoparticles (PNPs), nanofibers, nanocomposites have proven a superior efficiency in DD potentials such as food safety, pharmaceuticals, tissue engineering and regenerative medicine [2]. They have a profound capability of improving drug localization, solubility and allowing controlled release of encapsulated payloads. Nevertheless, the naturally occurring polymers have some limitations which curtails their application in DD products such as low mechanical strength, hydrophobicity, rapid rate of degradation, poor thermal stability and brittleness [3].

To amend this, studies are being conducted to surmount such shortcomings through combination with other synthetic polymers [3,4]. However, the irreplaceable merits of different classes of natural polymers over synthetic as drug carriers still urge researchers to depend solely on biomaterials (Figure 1A). These advantages include hydrophilicity, biocompatibility, non-immunogenicity, non-toxicity, non-toxic degradation products, antimicrobial and antioxidant capacities and high stability for tissue engineering [3]. From an environmental perspective, the worldwide demand for eco-friendly products has witnessed a sustainable growth for accommodating green technology advancements. Hence, industrial channels are searching for more reliable products to gain customers’ satisfaction and ease their commercialization [5].

On another avenue, post-synthetic modification of natural polymers seems an impressive way for ameliorating their functionalities, physical properties and broadening their potentials in DD. To date, natural polymers are still unable to stand alone or invade the market of food safety or biomedical products [6]. Post-synthetic modifications including blending, coating, electro-spinning, and plasma treatment, modifications of the polymer molecules by in vivo and in vitro enzyme treatment or chemical modification like epoxidation, hydroxylation, and carboxylation can be introduced to bring them to real-world applications [7]. Some studies have investigated naturally resourced polymers modification [8,9]; however, the application of these approaches for DD purposes is still in an immature stage. The main aim of this review is to provide an overview on the most recent investigations of antibacterial and antibiofilm polymeric NMs generated from natural polymers in eight different forms, namely PNPs, nanocrystals (NCs), liposomes, polymer based metal nanoparticles, dendrimers, micelles, nanocomposites and nanofibers (Figure 1B). Furthermore, the efficiency difference between free and polymer-based bioactive agents has been compared. The physiochemical characteristics, methodologies, polymer-polymer or drug-polymer interactions and potentials of recent antibacterial nanosystems have been tabulated. Moreover, the requirements, capacities, most practical and novel approaches of each form have been illustrated. The superiority of natural polymers over synthetic ones has been reviewed. More importantly, the post-synthetic modification of three polymers including polyhydroxyalkanoates (PHAs), chitosan and curdlan aiming at altering their structure and thereby their features have been displayed. Also, the review shads light on how chemical and enzymatic polymer modification could be a potential approach in developing sustainable, durable and efficient DD vehicles along with future outlook and opportunities.

## 2. Types of Polymeric NMs

In many parts of the world, medical, pharmaceutical and food industries have grown significantly, however these manufacturing processes are becoming unsustainable. Nanomaterial (NM)-based techniques have been shown in studies to have the ability to provide sustainable production alternatives [10,11]. Metal oxides, metal, magnetic materials, composites and ceramics and are only few examples of nanoparticles (NPs). These types of nanofabricates may not be fit for some industries. Thus, biopolymeric-based NMs represent a significant compatible replacement.

Polymer-based NMs can be classified as either natural or synthetic. Plants, animals, microbes, agricultural or animal wastes, and other biological sources can all be used to make natural or biopolymers. Proteins and carbohydrates are two types of natural polymers. Albumin, collagen, gelatin, fibrin, and zein are some of the proteins that have been employed to make NMs [12]. While chitosan (Cs), chitin, starch, cellulose, alginate, pectin, dextran and inulin are all common carbohydrates utilized in NM formulation. Polyesters such as polylactic acid (PLA), polyethers such as polyethylene glycol (PEG) and polyamides such as polyacrylamide are three types of synthetic polymers used to make NMs [13].

### 2.1. Particulate Nanodelivery

#### 2.1.1. Polymeric Nanoparticles

Polymeric nanoparticles (PNPs) have been extensively used in medicine, biotechnology, biomedicine, bioengineering, environmental technology, pollution control, conductive materials, sensors, electronics and photonics and are just some of the fields in which PNPs are employed [14,15]. PNPs in particular, can be used as a potential drug delivery vehicle, carrying medications to target cells and organs effectively [16]. Targeted delivery, better bioavailability, extended drug release, improved patient compliance, and fewer adverse effects are only a few of the benefits of nano-sized drug delivery systems [17]. Varying the mix of polymers, medicines, and other chemicals employed in the formulation can readily customize the surface properties and particle size of NPs [18,19]. Intestinal absorption and in vivo distribution are also influenced by particle size (Table 1) [20]. Using a rat model research, Panyam and his colleagues revealed that particles smaller than 100 nm appeared to give 15 to 250 times more cellular absorption than micro particles [21]. NPs have a large surface area, which assures a large payload [22].

Natural polymers are preferred over synthetic materials as drug carriers because they have various unique properties such as biocompatibility, hydrophilicity and non-toxicity [23]. Due to its low cost and availability, alginate (Alg) is one of the most favorable biopolymers among the several natural polymers [24]. The majority of alginate particles depicted in the literature are micrometer-sized [25,26]. Brown algae and bacteria produce sodium alginate, which is a natural anion branching polysaccharide polymer, made up of mannuronic acid (M block) and guluronic acid (G block) [27]. It is non-toxic, biodegradable, biocompatible and environmentally friendly, indicating that it has a lot of promise in the field of medication delivery [28,29].

The atomic economy and catalyst-free Ugi reaction were recently used to effectively develop and prepare a new alginate-based luminous polymer. Self-assembled luminous polymer NPs could be made from this amphiphilic polymer. With the addition of the aggregation-induced emission (AIE)-active material, alginate-based fluorescent polymer nanoparticles (Alg-TPVA) gained a fluorescence feature that showed clearly excitation-dependent [30]. The emission, which came in a variety of colors ranging from blue to red, demonstrated remarkable multicolor imaging capabilities. The synthesized Alg-TPVA exhibited stable micelle characteristics and can be utilized to deliver drugs in microenvironments. Furthermore, the fluorescent polymer nanoparticles (FPNs) have been effectively used in cell and plant leaf stomata to provide great multicolor imaging capacity. Water dispersibility, low critical aggregation concentration (CAC), and great biocompatibility have all been demonstrated with Alg-TPVA FPNs [30]. To promote the structural integrity and use, curcumin was chosen to be encapsulated in amphiphilic polymers. Alg-TPVA exhibits amphiphilicity and can self-assemble into spherical micelles in aqueous solution, whereas curcumin is a hydrophobic polyphenol [31]. Curcumin was encapsulated in an Alg-TPVA micelle and dialysis was used to test it under various pH settings. Due to the encapsulation of curcumin, the generated micelle demonstrated significant antibacterial activity and can readily pass through tight nozzles and channels due to its small particle size (125 ± 20 nm) [30]. This is attributed to the fact that curcumin has shown to possess potent anti-inflammatory, antibacterial and anticancer properties [32].

Furthermore, Thomas et al. [2] used a simple ionic gelation green process to make Rifampicin (RIF) loaded alginate NPs. Honey, which is known for its high biocompatibility and natural antibacterial agent, was used as a stabilizer in the preparation procedure [33]. Honey concentration is crucial for the creation of stable NPs. Honey formed clusters at concentrations below 1%, however, it maintained single droplets at higher concentrations [2]. Transmission electron microscopy (TEM), Fourier transform infrared spectroscopy (FTIR), Dynamic light scattering (DLS), and X-ray diffraction techniques were used to analyze the NPs’ physicochemical characteristics. The findings demonstrated that at concentrations of 1% (*w*/*v*) sodium alginate, 1% (*v*/*v*) honey, and 1% calcium chloride, clearly defined spherical Alg NPs were generated and selected for further investigations [2]. Drug encapsulation in nanosized particles has enormous promise in tuberculosis therapy because it improves drug intracellular availability. The intracellular environment causes the release of drug inside the cells once nanosized particles penetrate the bacterial cell wall [34]. The produced Alg NPs’ small size (64 ± 22 nm), Zeta potential (−4.72 ± 0.17 mV), drug EE (36.58% ± 2.15%) and high surface volume ratio increased the solubility of RIF, allowing for targeted delivery and making it a suitable carrier for RIF [2].

Several natural polyesters, such as PHAs, are also being studied as polymeric bases for nano manufacturing. PHAs, which are microbially created from renewable resources, may be the greatest replacement to traditional plastics [35]. Despite their enormous potential, commercial manufacturing of PHAs is still in its infancy [36]. Because of its biocompatibility and biodegradability, polyhydroxybutyrate-*co*-hydroxyvalerate (PHBV) has been established as a good polymer for drug delivery systems and bone tissue engineering. Chotchindakun et al. [37] inserted mesoporous bioactive glass NPs (MBGN) into PHBV to increase its bioactivity, while cinnamaldehyde (CIN) was loaded into MBGN to add antibacterial capacity. The emulsion solvent extraction/evaporation process was used to generate PHBV/MBGN/CIN microspheres. The average particle size and zeta potential of these microspheres, as well as their shape were examined. The average diameter was determined to be between 6.1 and 12.5 µm while the Zeta potential values were between −21.3 and −12.2 mV with high EE values ranging from 99.26% to 99.98%. This result might be explained on the basis of increased viscosity of the dispersed phase due to the higher polymer molar mass, which causes tiny microspheres to split more easily, letting large particles to develop [38]. The microspheres were loaded with different doses of CIN, resulting in PHBV/MBGN/CIN5, PHBV/MBGN/CIN10, and PHBV/MBGN/CIN20. In the first 3 h, these microspheres showed substantial antibacterial action against *Staphylococcus aureus* and *Escherichia coli*, with CIN releasing behavior lasting up to 7 days. The reason for the bactericidal action of CIN can be attributed to disrupting the synthesis of the cell membrane and ATP generation [39]. These results suggest that this system might be used as an alternate paradigm for an antibacterial biomaterial in bone tissue engineering [37].

El-malek et al. used the time kill method to assess the antibacterial potency of the free polyhydroxybutrate (PHB)-ASL10, PHB-ASL11 and their nanoforms against some reference strains (*Streptococcus pneumoniae* ATCC 6303, *S. aureus* ATCC 25,923 *Klebsiella pneumoniae* ATCC 35,627, *Pseudomonas aeruginosa* ATCC 27,853 and *E. coli* ATCC 25,922).

The free polymers had no bactericidal action [40], The polymeric NPs, whereas, demonstrated strong antibacterial effectiveness against gram-positive and gram-negative bacteria. The antibacterial prospect was clearly linked to the incubation time. This might explains why after 24 h of exposure to both polymeric NPs, all strains had the lowest absorbances. PHB-ASL10-NPs inhibited all bacterial strains, including *S. aureus* ATCC 25,923, which had the lowest OD_600 nm_ of 0.332. PHB-ASL11-NPs were shown to be antibacterial against *S. aureus* ATCC 25,923 [40]. PHB NPs exhibited a substantially higher inhibitory impact on bacterial cells development than free polymers at the same concentration. According to Liu et al. [41] polymers-NPs (PNPs) have a higher volume/surface ratio than free forms, which may explain their antibacterial superiority [42]. The direct contact and binding of the NPs to the peptidoglycan layer might be the likely mechanism behind the improved activity of PHB NPs. This causes an indentation on the bacterial surfaces as well as structural alterations, as well as exposing the underlying peptidoglycan layer through the disruption of the outer protective membrane. This type of injury frequently leaves the cell vulnerable, resulting in complete shrinking, seeping of cellular contents and cell death [43].

**Table 1 ijms-23-04101-t001:** Physichochemical properties of recently investigated bio based PNPs and their applications.

Polymer	Nanosystem	Particles Size Distribution	Encapsulation Efficiency	Preparation Method	Indication	Reference
Chitosan	Cs-NPs embedded gelatin nanofibers	From 154.9 ± 3.0 to 236.3 ± 2.6 nm	39.6 ± 0.8%	oil-in-water emulsion/ionic-gelation	Anti-biofilm against *E. coli* O157:H7 in food samples	[44]
	peppermint oil-loaded composite microcapsules with hydroxypropyl methyl cellulose/Cs/silica shells	118.12–152.5 nm	89.1%	Pickering emulsion	Prolonged (60 day) antibacterial agent in food preservation	[45]
	Gemifloxacin-loaded Cs-NPS	158.4 nm	46.6%	Ionic gelation	antibacterial ocular dosage form	[46]
	Carvacrol-loaded hydrophobic Cs-based NPs	134 ± 5 to 220 ± 12 nm	From 19.3 ± 4.28 to 56.4 ± 0.6%	Ionotropic gelation	A promising antibacterial biomaterial in health science and material engineering.	[47]
	Thymol loaded Cs-NPs	338.92 ± 8.30 nm	-	Oil-in-water emulsion	Long-term quality retention and decay inhibition of chestnut	[48]
	Cs-NPs loaded with β-lactam antibiotics and β-lactamase inhibitors	210 ± 13 to 240 ± 8 nm	46 to 56%	Water/oil/water microemulsion	Treatment of diseases caused by critically important ESBL-producing multidrug resistant pathogens	[49]
Cellulose	Bacteriocin immobilized crystalline nanocellulose	Whisker shaped particles having the length 71.2 ± 20.7 nm and width 27.8 ± 11.2 nm	-	Pressure homogenization	A potential antimicrobial packaging film with enhanced mechanical properties.	[50]
Starch	Starch NPs as vehicles for curcumin	135.1 to 190.2 nm	85 to 90%	Nanoprecipitation	Nutraceutical and drug delivery system	[51]
	Nano-encapsulated catechin in starch NPs	323 to 615 nm	48 to 57%	Nanoprecipitation	A bioactive ingredient in the functional foods	[52]
	Curcumin-Loaded Starch NPs	141 ± 12 to 198 ± 14 nm	82 to 92%	Nanoprecipitation	Controlled release of bioactive compounds in Highly hydrophilic/lipophilic foods	[53]
Curdlan	Cationic curdlan- anionic hydroxypropyl-cellulose	200 nm	94.80%	Immediate self-assembly	A new system as a promising carrier to deliver piroxicam efficiently and safely to the patients	[54]
Curdlan Cyclodextrin	Cyclodextrin conjugated curdlan NPs	187 ± 48 nm and 619 ± 176 nm	loading efficiencies of Rifambicin and Levofloxacin are 60 and 30 μg per mg of nanoparticle	Solvent evaporation	A promising system for the loading and intracellular release of hydrophobic drugs into macrophages for various therapeutic applications (Tuberculosis treatment)	[55]
Zein	Encapsulated pecan nut shell in zein microparticles	481 to 493 nm	23.7 to 59.9%	Microencapsulation	Delivering phenolic compounds with applications in functional foods and nutraceuticals	[56]
Lactoferrin (LF) Pectin	Pectin-LF nanocomplex loaded with curcumin	100 to 300 nm	85.3%	Nanoprecipitation	Nanoscale food grade delivery systems for the improved water solubility, controlled release, and antioxidant activity of hydrophobic curcumin	[57]
Cyclodextrin	Naringin Loaded β- Cyclodextrin NPs	70.00 ± 15.06 nm	93.54 ± 2.6%	Nanoprecipitation	Enhanced bactericidal potential against *E. coli, Listeria monocytogenes* and *E. faecalis*	[58]
LF Chondroitin	LF-chondroitin nanocomplex loaded with resveratrol-quercetin NCs	2333.5 ± 6.59 nm	85.2% Res. 90.1% Quer.	Self-assembly electrostatic interaction	Dual therapeutic targeting of bacterial pneumonia and lung cancer	[59]

#### 2.1.2. Polymer/Drug Nanocrystals (NCs)

Nanocrystallization seems a highly successful approach for improving the aqueous solubility of poorly water-soluble compounds. The compound’s surface area increases when it is reduced to nanoscale size, which accelerates dissolution and diffusion (absorption) correspondingly (Figure 2A). Parent-compound nanoscopic crystals having a diameter of less than 1 mm are referred to be drug nanocrystals. They are made entirely of non-carrier drug and are often stabilized with polymeric steric stabilizers [60]. Furthermore, homogenization and sonication of NCs are thought to be effective methods for decreasing the size of NMs, triggering better dissemination, and lowering aggregation and nucleation. To control particle size and shape, Jarvis et al. [61] proposed a temperature-controlled antisolvent precipitation approach with suitable stabilizers. On another avenue, hybrids of proteins, polysaccharides and other natural polymers play a crucial role in developing targeted DD systems in terms of controlling NCs size, allowing multiple drug loading and synergy, avoiding particles size growth, obtaining sustained release and boosting the system stability [62].

In this regard, a recent research project in our laboratory has succeeded in fabricating LF-chondroitin electrostatic nanocomplex loaded with resveratrol and quercetin NCs with EE of 85.2% and 90.1% *w*/*w*, respectively. The system also displayed an impressive physicochemical stability over a period of 12 weeks. The authors have postulated that the polymeric matrix could have protected the loaded drugs (resveratrol and quercetin) from degradation, displaying a satisfactory stability over along time of storage in the colloidal state. The developed nanosystem (233.5 ± 6.59 nm) had a remarkable antibacterial attribute in treating mice with polymicrobial lung infection, resulting in a significant reduction in *S. aureus* and *K. pneumoniae* bacterial counts as compared to free resveratrol and quercetin alone. The superiority of the polymeric NPs over the free drugs is corelated to the high EE of the loaded compounds and the sustained release of the bioactive agents over 48 h, which preserve the therapeutic concentration in the body for longer periods, resulting in more bacterial count reduction and clearance [63]. The safety of the NPs was affirmed by assessing liver and kidney functions of mice after administration, which could be explained on the fact that all the utilized polymer (LF and chondroitin) and drugs (resveratrol and quercetin) are naturally sourced and FDA-approved. Thus, their biodegradability and metabolism in the body are not worrying [59].

In the same context, Tomić et al. [64] increased the solubility and the antibacterial capability of azelaic acid by synthesizing azelaic acid NCs (58.8 ± 2.5 nm), increasing its cutaneous bioavailability via loading them within Pluronic F127/hyaluronic acid hydrogel matrix. Moreover, an impressive combination of PHB and Cs was proposed in a recent research in our lab to create a functional blend of for mitigating wound infection. To overcome PHB’s inherent shortcomings, such as brittleness, thermal instability in the molten state, sluggish rate of degradation and acidic degradation, it was mixed with Cs, with the hypothesis that both natural polymers have mutually beneficial effects. To improve drug transdermal administration, water insoluble kaempferol (KPF) was pre-formulated into water soluble KPF-NCs with 451 nm particle size and high colloidal stability (310.1 mV) with 96.6% EE. The film possessed excellent breathability, thermal stability, and mechanical strength (331 MPa). The integration of KPF-NCs and increasing the ratio of Cs was proven to enhance the psychochemical properties of the blend, reducing its crystallinity (Figure 2B). The improved flexibility and physicochemical properties of the blend could be attributed to the self-assembly of PHB (hydroxyl groups) and Cs (amine groups) resulting in better crosslinking and reticulate structure. The addition of KPF-NCs enhanced the number of binding sites between Cs and the NCs, resulting in a dense and stable matrix [65]. Plant-derived compounds, in reality, not only offer action (antioxidant and antibacterial), but also serve as cross-linking agents [66].

To enhance its antibacterial activity against pathogenic bacteria, the particle size of KPF-NCs was lowered inside a PHB-Cs film matrix. This is because lowering particle size increases their ability to enter and concentrate in bacterial cells, increasing their availability and effectiveness. Another theory is that the particles’ tiny size allows them to generate gaps in the peptidoglycan-based bacterial cell walls, resulting in cell death [67]. According to the authors, the size of the generated KPF-NCs was deemed appropriate for causing cell disintegration while demonstrating substantial antibacterial effectiveness after 2 days of controlled release. That is why, Cs+KPF-NCs displayed a synergetic antibacterial capacity against multidrug resistant (MDR) *S. aureus* and *Acinetobacter baumannii* with nearly 100% cell viability decrease after 48 h, which was verified by SEM, demonstrating the significant impact on bacterial morphology and structural integrity [68].

On another avenue, polymers also can be transformed to NCs with the purpose of boosting their mechanical potentials. Among naturally occurring polymers, nanocellulose, including cellulose nanocrystals (CNCs) is deemed an appealing NM to improve polymer qualities because of its good physicochemical features, such as large surface area, superior optical, mechanical, and biocompatibility, NM is used [69]. Moreover, due to the presence of hydroxyl groups and the increased surface area of nanocellulose, greater interactions with diverse chemical moieties are possible. CNCs, also known as cellulose nanowhiskers, have a highly crystalline structure and a standard width of 10–20 nm and length of 100–250 nm. Patel et al. [65] has created a multi-functional Cs/cellulose NCs hydrogel scaffold with improved mechanical strength and cytocompatibility. The modulus of the composite scaffolds was greater than that of the pure polymer scaffold (determined by the slope of the initial linear section). The existence of crystalline CNCs within polymer chains, which permits load transfer from the polymer to the CNCs via stronger connections during measurement, was ascribed to the higher modulus values (14.6–87.0 KPa) in the composite scaffolds. To get the best reinforcing effect from the NM, better interfacial contact between the polymer matrix and the included NM is crucial [70]. The presence of hydrophilic CNCs and their surface shape contribute to the composite scaffolds’ enhanced biocompatibility. The porous and structured morphological characteristics of the composite scaffolds were visible in SEM images, increasing cell survival by giving a larger surface area to the developing cells. Surface chemistry, roughness, and stiffness of materials all have a significant impact on the cellular activity. When compared to pure polymer scaffolds, NM-based scaffolds displayed enhanced cell survival, proliferation and differentiation due to their superior topographical surface chemistry [71].

The hydrogel scaffold demonstrated significant antibacterial attribute and sustained release of tetracycline hydrochloride. Scaffolds containing 4% CNCs outperformed others in terms of antibacterial potential. The charged groups in the composite scaffolds are thought to have harmed the bacterial cell membrane by increasing electrostatic interactions between the amino groups of Cs and the phospholipid components of bacterial cell membranes. This finding suggested that the manufactured scaffolds had better antibacterial capability and might be used as antibacterial agents to inhibit bacterial proliferation [65]. Because of the presence of the crystalline CNC NM, which delayed the release of the loaded drug, prolonged drug release was seen from the composite scaffolds compared to the pure polymer scaffold. By virtue of these merits, the manufactured scaffolds had a high potential for use as promising biomaterials in a variety of applications [65].

#### 2.1.3. Polymer-Based Metal NPs

In recent decades, metal-based NPs (MNPs) have been presented as effective antibacterial alternatives to antibiotics [72]. MNPs’ unique properties, such as ultra-small dimensions and huge surface area, make them ideal for a wide range of therapeutic applications, including cancer therapy, microbial disease treatment and biofilm inhibition. MNPs such as Au, Ag, CuO Fe_3_O_4_, Se, TiO_2_ and ZnO have been proven as antibacterial coating agents. However, the cytotoxic impact of MNPs is still concerning many researchers when it comes to adjusting treatment doses and examining liver and kidney profiles. MNPs synthesis generally consists of three steps: metal ion reduction, nucleation, and subsequent stabilization [73]. The content of the stabilizer utilized is directly connected to the synthesis’s efficacy as well as the form of the finished NPs. Green MNPs synthesis has recently received a lot of attention in comparison to chemical and physical techniques due to its environmental and economic benefits [74]. Biological sources such as bacteria or fungi, natural plant extracts or natural stabilizers might reduce inorganic metal ions to MNPs. Among these approaches, polymeric matrices seem one of the most convenient ways for MNPs synthesis and carrying (Table 2) [75]. To illustrate, polymeric matrix alleviates MNPs toxicity by controlling the particles release, providing a therapeutic concentration while maintaining safety. Thus, polymer-based MNPs platforms have been widely used in in vivo studies [75].

In this context, silver NPs were synthesized by using polymer/protein mixers of cashew gum-hydrolyzed collagen, carrageen anhydrolyzed collagen, and agar-hydrolyzed collagen [75]. To prepare AgNPcolCarr and AgNPcolAgar, homogenized mixes of collagen, aqueous cashew gum and carrageenan or agar were utilized. It was noted that all AgNPs samples have average particle diameters of less than 50 nm, demonstrating the efficiency of these polymer mixtures in MNPs generation. The AgNPcolCarr sample had minimum inhibitory concentration (MIC) values of 62.5 and 31.25 µM/mL against *E. coli* and *Salmonella* sp., respectively. The data support the fact that the size of the particle can have a significant impact on the antibacterial mode of action. AgNPs offer a broad surface area for bacterial interaction, allowing the particles to cling to the cell membrane causing cell membrane damage [76]. Also, Ag^+^ causes the generation of reactive oxygen species (ROS), such as hydroxyl radicals (•OH), which can cause cell death in gram-negative bacteria by increasing membrane permeability. Furthermore, AgNPcolAgar and AgNPcolCashew were both 2.0 and 6.5 times less cytotoxic than AgNO_3_ which could be owing to the fact that AgNPs have been reduced and capped by natural polymeric mixtures [75].

In addition, in situ synthesis of MNPs in biopolymer matrices is a unique approach for the fabrication of bionanocomposites with superior NPs dispersion, which have promising applications in environmental and biomedical fields. Muthulakshmi et al. [77] has proposed a novel green sustainable bionanocomposites comprising cellulose and copper NPs (CuNPs) by utilizing a bioflocculant as a copper ion precursor reductant for the in situ CuNPs synthesis in the matrix of cellulose. Cellulose wet sheets were immersed in various concentrations of aqueous copper sulfate precursor solutions. The precursor solution permeates the matrix wet films and, upon contact with the diffused bioflocculant, reduces to CuNPs. The presence of several functional groups (glycoprotein) in the microbial polymeric bioflocculant helps to reduce the size. It is worth noting that, CuNPs in the 100 nm size range were uniformly distributed throughout the polymer matrix [77]. The cellulose biopolymer served as an appropriate matrix for the stabilization of the CuNPs within the cellulose sheets. The NPs exhibited a high surface-to-volume ratio and the uniform distribution of NPs in the matrix promoted contact between the polymer chain and the NP surfaces, improving the antibacterial efficacy of the composites, especially at greater CuNPs concentration in the matrix. The inhibitory potential of cellulose/CuNPs bionanocomposite films and CuNPs produced outside the matrix at 200 g/mL and 500 g/mL. It was recorded that the cellulose/(250 mM) bionanocomposite has the largest inhibition zone (IZ) of 12 mm against *E. coli*, while the cellulose/CuNP (125 mM) bionanocomposite film had a 9 mm IZ and the cellulose/CuNPs (5 mM) and cellulose/CuNPs (25 mM) did not show any zone of clearance. The concentration of NPs present inside the cellulose matrix was directly related to the bacterial strain’s zone of clearance. Copper ions produced by flaws in copper NPs contribute to the bactericidal actions. In fact, cupper metal ions could have blocked the protein’s functional groups and alter membrane integrity [78]. Hence, owing to the cost effectiveness, high biodegradability, potent antibacterial capacity, CuNPs developed by in situ generation in polymeric matrix are a promising candidate in food safety and biomedical potentials [79].

**Table 2 ijms-23-04101-t002:** Different forms of polymeric -based MNPs and their applications.

MNPs	System	Particle Size	Application	Reference
AgNPs and Ag/TiO_2_	Cs-PVC conjugates encompassing Ag NPs	3–7 and 15–22 nm	A revolutionary antibacterial material that can be useful in biomedical applications	[80]
AgNPs	Silver NP stabilized by hydrolyzed collagen and natural polymers	15.82 ± 10.82 to 46.42 ± 6.22 nm	Natural polymeric blends loaded with AgNPs as antibacterial antifungal biomaterials	[75]
AuNPs	Gold NPs/polyaniline boronic acid/sodium alginate aqueous nanocomposite	35.2 ± 1.3 to 64.8 ± 2.4 nm	Nanocomposite based on chemical oxidative polymerization for biomedical application	[81]
Nio/NiONPs	Green nickel/nickel oxide NPs	48.6 ± 0.9 nm	NPs for prospective antibacterial and environmental remediation applications	[82]
CuNPs	Cellulose/copper NPs bionanocomposite films	20–80 nm	A novel cellulose bionanocomposites appropriate for food packaging as well as corrosion resistant coating applications	[77]
AgNPs	Cs-stabilized silver-colloidal NPs immobilized on white-silica-gel beads	20–30 nm	A promising composite as antibacterial air filter	[83]
AgNPs	Polymer encapsulated silver nanoparticle coatings	~2 nm	Coatings materials material in drinking water purification, air filtration, domestic and Industrial air quality management and antibacterial packaging	[84]
AgNPs	Polymer mediated synthesis of cationic silver NPs	4.99 ± 2 to 29.88 ± 5 nm	Hybrid NPs with promising applications against antifungal-resistant microorganisms	[85]
AgNPs	AgNPs–Polymer Composites	20–25 nm	Nanoparticle–Polymer Composites to protect patients from nosocomial infections and prevent biofilm development over time.	[86]
CuO NPs	Various polydopamine -modified CuO NPs by hydrothermal synthesis	21.79 to 24.84 nm	Polymer -modified CuO NPs antibacterial agent for usage in biomedical and food preservation applications	[87]
AgNPs	Microwave-Assisted Silver NPs Decoration and Polymer-Based Graphene Derivatives	35 nm	A suitable alternative to traditional antibacterial agents, which cause leaks in the environment and/or in the tissue of living beings	[88]
AgNPs	Alginate-based biodegradable films containing silver NPs and lemongrass essential oil	5 to 25 nm	Biodegradable alginate films are a new type of biodegradable film that may be used to monitor the storage conditions of hypersensitive foods.	[89]
ZnONPs	ZnO nanoparticle-enhanced alginate films with citronella essential oil (CEO)	48–89 nm	A possible antimicrobial wrapping for cheese and other foods	[90]

#### 2.1.4. Liposomes

Liposomes are known as spherical vesicles with at least one lipid bilayer. They are mostly made up of phospholipids, primarily phosphatidylcholine, but they may also contain other lipids like phosphatidylethanolamine as long as they don’t interfere with the lipid bilayer structure [91]. Surface ligands may be used in a liposome design to adhere to diseased tissue. They can also be used to transport nutrition and pharmacological treatments, such as lipid nanoparticles in mRNA vaccines and DNA vaccinations. Disrupting biological membranes by sonication can be used to generate liposomes [91]. The four primary forms of liposomes are the cochleate vesicle, the small unilamellar liposome vesicle (SUV, only one lipid bilayer) and the large unilamellar vesicle (LUV). The Fourth type is the less desirable type, the multilamellar vesicle (MLV, with several lamellar phase lipid bilayers) in which one vesicle includes one or more tiny vesicles [92].

Liposomes’ basic characteristics are primarily expressed in the following aspects; good biocompatibility, sustained release, reduced diffusion rate of core material, excellent dispersibility in water, and long residual action due to their unique molecular structure and physicochemical properties [93]. A hydrophilic core and one or more phospholipid bilayer membranes comprise liposomes. As a result, liposome encapsulation might be a potential strategy for improving bioactive compounds’ stability and bioavailability. Liposomes, on the other hand, tend to agglomerate, fuse, or disintegrate due to their soft material properties [94].

Recently, research on liposomes containing essential oils is gaining traction [95]. The primary component of clove oil, eugenol, was discovered to be efficiently preserved by liposomes [96]. Clove oil liposomes were added to soybean products, demonstrating that they had remarkable antibacterial effects against *S. aureus* [97]. Cinnamon essential oil and cinnamaldehyde were encapsulated in egg yolk lecithin for use in edible film, which successfully sustained essential oil release and boosted the antibacterial characteristics’ durability [98]. More recently, liposomes were used to encapsulate cinnamonaldehyde and the impacts of the core-wall ratio on storage stability and antibacterial properties were investigated. Particle diameter during storage indicated that cinnamaldehyde liposomes with a high core-wall ratio amalgamated more easily, with a retention ratio of about 60% [99]. The flexibility of the liposome membrane may be reduced when cinnamaldehyde levels rise. As the core-wall ratio rises, the average particle size rises as well. Because lipid membranes with a slight curvature were less fluid, increasing the core-wall ratio can help minimize core material leakage to some extent [100]. Furthermore, fluorescent labeling and the death log value were used to study the antibacterial activity of cinnamaldehyde liposomes against *S. aureus* during storage [99]. The killing log value of pure cinnamaldehyde liposomes treatment during 24 h before accelerated storage at 37 °C was much greater than that of cinnamaldehyde liposomes. This was attributable to the fact that when the core-wall ratio increased, liposome EE decreased, and more free cinnamaldehyde was exposed to bacteria, resulting in more effective bacteriostatic activity [99]. After storage, liposome-encapsulated cinnamaldehyde proved to be more effective than pure cinnamaldehyde at suppressing bacteria by compromising cell membrane integrity. As a result, liposomes may aid in the long-term stability and antibacterial activity of cinnamaldehyde.

Similarly, gallic acid liposomes (GA-LIP) and LF-decorated gallic acid liposomes (LF-GA-LIP) were created. LF-GA-LIP had a greater average particle size (153.2 ± 1.4 nm) than GA-LIP, and the former had better EE and storage stability. GA-LIP containing 2% gallic acid showed a greater EE and smaller particle size [101]. Gallic acid’s interaction with the polar headgroup of phospholipid in the bilayer resulted in its spanning orientation in the liposomal membrane, potentially increasing liposome volume and size [102]. However, increasing the gallic acid loading percent to 8% did not result in higher particle sizes. This phenomena might be reasoned to gallic acid saturation in the vesicles, as evidenced by the fact that the EEs of GA-LIP fell dramatically when the loading quantity was raised from 4% to 8% [103]. TEM and atomic force microscopy revealed the spherical formations of both liposomes and that Gallic acid was found in an amorphous form. In simulated digestion, LF-GA-LIP showed a delayed-release impact when compared to GA-LIP [101]. The swelling and clumping of liposomes in simulated saliva might explain the rise in particles size. The aggregation of liposomes might be caused by the salt and polyelectrolyte bridging activity, as well as the binding behavior of mucin to liposomes in saliva [104]. Additionally, loaded LF-GA-LIP exhibit better antibacterial activities than GA-LIP against *E. coli* and *S. aureus*. The MICs of GA-LIP and LF-GA-LIP against *S. aureus* and *E. coli* were lower than those of LF and GA, indicating that GA-LIP had enhanced antibacterial activities. The diameters of IZs of LF-GA-LIP were bigger than those of GA-LIP for both strains at the same concentration, suggesting that LF-GA-LIP had superior antibacterial characteristics than GA-LIP, which might be partly owing to LF’s antibacterial action [105]. LF antibacterial activity is related to Fe^3+^ ion absorption, which limits bacteria’s usage at the afflicted organ, decreasing their growth and alleviating their pathogenicity [59]. The findings displayed that liposomes coated with LF may be developed as a promising delivery technology for use in the food sector.

In order to target intra-monocyte Methicillin-resistant *S. aureus* (MRSA) infections, gelatinized core liposomes (GCL) were employed to increase the encapsulation and antibacterial action of the novel natural hydrophilic antibiotic kojic acid (KA). The antibacterial action was tested against MRSA using spectrophotometric microdilution [106]. The Histopaque separation was employed to measure the quantitative absorption of the formulations in monocytes. The formulations were also put through the MTT assay to see if they were cytotoxic to monocytes. Finally, gelatinized liposomes containing 80 mg gelatin (GL-80) were used to treat intra-monocyte MRSA infection. With a mean size of 188.8 ± 11.5 nm, the GL-80 formulation dramatically boosted EE by 50%. Clearly, the inclusion of KA may be the reason for that the simple formulations have a significant size decrease when compared to the loaded formulas. Furthermore, because the formulations’ size distribution was greater than 150 nm, they were classified as vesicles, which is essential for monocyte/macrophage recognition and phagocytosis. This observed vesicular diameter is also important for the vesicles to have an enhanced penetration and retention (EPR) impact in inflamed tissues [107] with less clearance from the body [108]. The EE witnessed a considerable rise when the amount of gelatin in GL-80 and GL-40 was increased. This might be due to an increase in the viscosity of liposomes’ vesicular core, as well as negatively charged kojic acid and positively charged type A gelatin are likely to interact electrostatically and via hydrogen bonds [109]. The GL-80 and conventional liposomes (CL) formulations had the same MIC value of 6 mg/mL. The GL-80, on the other hand, demonstrated a greater percent absorption by monocytes with almost no cytotoxicity. Furthermore, a considerable decline in intracellular MRSA infecting monocytes was achieved over time. At all doses from 25 to 300 g/mL, GL-80, CL, and free KA demonstrated adequate cell vitality, with percent cell viability exceeding 80%. GL-80 can deliver the desired therapeutic benefit with lower dosages, lowering the risk of side effects and toxicities. The uptake of KA by monocytes from liposomes might be mediated by phagocytosis due to the vesicular size range of formulations [110] or by cellular enzymes inside the monocytes degrading the liposomes and leaking the KA. The KA-loaded GCL is a viable platform for targeting MRSA infection of inflamed tissues inside monocytes. Antibiotic resistance was combated by using an unique nano-carrier to enhance the action of a new natural antibiotic substance [106].

#### 2.1.5. Micelles

Micelles are colloidal suspensions made up of supramolecular assemblies or aggregates of surfactant phospholipid molecules distributed in a liquid. In water, a typical micelle aggregates with the hydrophilic portions in contact with the surrounding solvent and the hydrophobic single-tail sections sequestered in the micelle center [111]. The packing characteristic of single-tail lipids in a bilayer causes this phase. The creation of the micelle is caused by the difficulties of filling the whole volume of the interior of a bilayer with tolerating the area per head group placed on the molecule by lipid head group hydration [112]. A normal-phase micelle is the name for this sort of micelle (oil-in-water micelle). The head groups are at the center of inverse micelles, with the tails spreading outward (water-in-oil micelle) [112]. Micelles have a roughly spherical form, while, other phases are also feasible, such as ellipsoids, cylinders, and bilayers [113]. A micelle’s form and size are determined by the molecular geometry of its surfactant molecules as well as solution variables including ionic strength, pH, surfactant concentration and temperature [113]. Micellisation is the process of creating micelles and it is part of the phase behavior of several lipids depending on their polymorphism [113].

According to reports, the incorporation of nanocarriers such as micelles served as vehicles for transporting charge-weak medicines into multifunctional coatings [114]. Nonetheless, these approaches are limited in practical applications due to a number of issues such as complex synthesis and poor loading efficiency. As a result, there is a pressing need for the invention of a diverse approach for loading weakly charged antibiotics and drugs into coatings using commercially accessible building blocks [115]. A unique and simple method for generating pH-responsive layer-by-layer (LbL) films implanted with polymeric micelles as nano-vehicles loaded with charge-weak antimicrobial medicines was devised, allowing efficient drug loading. Tobramycin (Tob)-embedded heparin miscells (HET) and positively charged chitosan (CHT) were used as pH-responsive LBL multilayer building blocks, respectively [116]. The morphologies, chemical compositions and hydrophilicity of the transformed surface were characterized, confirming the successful deposition of the Tob-loaded CHT/HET multilayers coatings on the polydopamine-modified Titanium surface. The drug release profiles showed quick release at pH 7.4 and sluggish release after being exposed to mildly acidic conditions [116]. A rough surface of polished Ti substrate was plainly apparent, with precise and perpendicular scratches that turned somewhat glossy following deposition of poly dopamine layer and grew more uniform and smooth after immobilization of (CHT/HET)_8_ multilayers. It was claimed that there were two processes involved in the formation of CHT/heparin PEMs: The first phase was the creation of an island-like structure on the surface, which was then subsequently replaced in the second phase by a step-shaped arrangement [117]. Antibacterial studies revealed that the Tob embed CHT/HET nanostructured multilayers not only substantially prevented initial bacterial adherence but also interfered with biofilm development. In particular, in acid conditions these functional coatings demonstrated a “long-term antibacterial” pattern. All of the antibacterial coating building blocks, CHT, heparin and Tob are widely used in the field of biomaterial implants due to their excellent biocompatibility and ability to inhibit microbial colonization [118]. Heparin with a strong negative charge reduces bacterial adhesion and biofilm formation by rejecting negatively charged pathogens, despite the fact that the antibacterial advantage of heparin was practically negligible in this research due to micelle production consuming most of the negative charges of heparin [119]. In a brief, These multilayer coverings loaded with a variety of medicines hold great promise in lowering postoperative infection and meeting an unmet therapeutic need [116].

Another study attempted to create a hydroxyapatite (HAP)-based scaffold composite for orthopedic applications using a Casein (CAs) micelle Considering the production and analysis of several fluorine (2% and 5%) substituted HAPs (FHAP), they were examined for drug release and antibacterial effectiveness of Ciprofloxacin (CIP) [120]. The effective production of the HAP composites is confirmed by physicochemical characterization such as FTIR and Raman spectroscopy. Powder XRD and FESEM analyses were also utilized to establish crystallinity and morphological behavior, respectively. EDX analysis was used to confirm the elemental makeup. The Raman spectra shows a prominent characteristic peak of the tetrahedral PO_4_ group at 962 cm^−1^ as the main one, which corresponds to the stretching of the symmetric P–O bond. The other PO_4_ peaks were also resolved and ascribed to phosphate groups’ intrinsic vibrational modes [121]. The antimicrobial investigations show that the 5% FHAP sample has better antifungal and antibacterial properties, with the maximum activity reported against gram-positive bacteria (*S. aureus*) with an IZ of 47 mm and the gram-negative bacteria (*E. coli*) with an IZ of 38 mm. With the CAs/5% FHAP sample, the CIP medication release profile has been controlled. As a result, this composite was created for scaffold construction using CAs-alginate matrices. Furthermore, analysis of the CAs-alginate/5% FHAP scaffold composite reveals that it is porous, biodegradable, has significant water absorption and retention ability, and has regulated CIP drug-releasing capabilities [120]. According to these findings, the synthesized CAs-alginate/5% FHAP scaffold composite may be appropriate for biomedical and bioengineering applications of bone tissue formation and as an implant.

More recently, for the first time, unique shell nanostructures of Copper-metal organic frame-work (Cu-MOF) compound were manufactured using an effective, rapid and controlled approach of ultrasonic aided reverse micelle [122]. The Fe_3_O_4_ nanoparticle was utilized as the core to improve the physicochemical characteristics of these compounds as well as their stability. The findings demonstrated that Fe_3_O_4_@Cu-MOF/core-shell nanostructures have significantly greater thermal stability than pure Cu-MOF samples. This difference is due to the structural stability of Fe_3_O_4_@Cu-MOF/core-shell nanostructures. Furthermore, the DSC findings for both samples are identical and show the amount of change in the necessary energy for weight loss at each step [123]. The antibacterial findings of the inhibition assay zone demonstrate that Fe_3_O_4_@Cu-MOF/core-shell is more effective against *E. coli* and *S. aureus* than Fe_3_O_4_ or Cu-MOF alone. Because of their size, nature and internal surface volume, Cu-MOF and Fe_3_O_4_@Cu-MOF/core-shell MOFs can be employed as reservoirs for the antibacterial aggregate rather than Fe_3_O_4_. MOFs are employed to progressively release metal ions to provide long antibacterial durability. Although the Fe_3_O_4_ in Fe_3_O_4_@Cu-MOF/core-shell does not have significant antibacterial capabilities, it can stabilize the Cu-MOF/core shell and generate persistent copper ion release [122]. Interestingly, the concentration of the Fe_3_O_4_@Cu-MOF/core-shell nanostructure increases its antibacterial activity. This can be linked to the bacteria’s cellular structure. The cause for this behavior is the variation in cell wall structure between these two bacteria. Gram positive bacteria’s cell wall was naturally generated by the peptidoglycan structure, which made up 80% of the cell wall. This structure resembles only 10% of the cell wall of the gram-negative bacteria and lipopolysaccharides, phospholipid, and lipoprotein make up 50%, 35% and 15% of the exterior membrane, respectively [124]. The produced Fe_3_O_4_@Cu-MOF/core-shell nanostructures have demonstrated reasonable antibacterial activity against both gram positive and gram negative microorganisms [122].

#### 2.1.6. Dendrimers

Dendrimers are one of the innovative drug delivery technologies based on nanotechnology that have piqued the interest of many drug delivery researchers. Such three-dimensional polymeric nanostructures have uniform size and surface functionalization, making them an appealing tool for biological applications [125]. Dendrimers are tree-like, highly branching polymeric architectures with varied biopharmaceutical features such as high surface functionality density, good solubility, well-defined nanostructure, low immunogenicity and monodispersity. These properties drew a lot of interest to dendrimers in a variety of domains, particularly in the health sciences [126]. This technology might be used to transfer both hydrophobic and hydrophilic components, such as genetic and imaging molecules. This delivery technology is now being used to treat a variety of ailments in order to overcome the limitations of traditional delivery methods [125].

Antimicrobial peptide dendrimers (AMPDs) are newly created nanostructures with a broader range and more antibacterial activity than antimicrobial peptides (AMPs) [127]. Several approaches have been suggested to enhance antimicrobial drug activity while decreasing in vivo degradation and toxicity [128,129]. Peptide PEGylation may prolong serum half-life, while the in carriers-peptide encapsulation could hinders the proteolytic destruction and peptide sequence optimization may aid to optimize the efficacy/toxicity balance. Nonetheless, these improvements have resulted in little to no clinical translation. Throughout this scenario, the linkage of AMPs to polymers may offer possibilities for protecting the peptide from degradation and/or sustaining its activity at the disease spot [130,131]. Patrulea et al. [132] have created a novel chemical platform for connecting Cs derivatives to AMPDs of varying ramification and molecular weights using thiol-maleimide reactions. They demonstrated that combining AMPDs with Cs derivatives allowed the two drugs to work synergistically. When AMPD conjugates were included into various biopolymer formulations, such as NPs, gels and foams, the antibacterial action was sustained [132]. Increasing the crosslinker’s alkyl chain from propyl to decyl, or by adding additional functional groups to the existing 6 carbon spacer linked to Cs derivatives had no effect on antibacterial efficacy [133]. Importantly, the antibacterial activity of native G3KLcys peptide (MIC: 16 g/mL) is maintained during the interaction with Cs derivatives via any of the crosslinkers used. Surprisingly, the authors discovered that the conjugates had much stronger antibacterial activity than the native AMPD for all crosslinkers examined [132]. Using electron and time-lapse imaging to investigate their mode of action, the researchers discovered that the AMPD-Cs conjugates were absorbed after disrupting the bacterial inner and outer membranes. The authors also demonstrated that AMPD conjugates are not harmful to mammalian cells. G3KLcys-Cs conjugates significantly damage both inner and outer bacterial membranes, as revealed by TEM and SEM imaging. Using confocal time-lapse imaging, we next followed the conjugate’s transit in *P. aeruginosa* across time. In the presence of propidium iodide (PI), a fluorescent dye that binds to DNA, bacterial cells (10^9^ CFU/mL) were incubated, after bacteria are permeabilized, with Fluo-G3KLcys-DAH-CMTMC at 40 g/L (equal to 2 MIC). Bacterial cells were confined inside an M63 agarose pad for live imaging, then PI was added, recording began, and Fluo-AMPD-conjugate was added [134,135]. This chemical technical platform might be utilized to create novel membrane disrupting treatments to eliminate microorganisms found in acute and chronic wounds.

Similarly, owing to their self-assembly and unusual structure, resorcinarene-based macrocycles have received a lot of interest in the field of drug administration. A recently published study is centered on the development of a novel resorcinarene-based macrocycle (Benzyloxy Macrocycle, BM) to improve the therapeutic effectiveness of quercetin (QRT) [136]. FT-IR, Mass, ^1^H NMR, ^13^C NMR spectroscopy were used to characterize the synthesized BM, while cytotoxicity and hemolysis tests were used to assess its biocompatibility. The findings supported the chair confirmation of synthesized BM which has previously been documented in the literature [137]. BM’s FT-IR spectra revealed hydroxyl stretching at 3386.6 cm^−1^ and CH_2_ stretching at 2930.7 cm^−1^. Aromatic C=C was detected at 1607 cm^−1^ and –COC– functionality was detected at 1239.4 cm^−1^. All of the spectrum discoveries support the effective synthesis of BM. The anti-bacterial efficacy of QRT-loaded dendrimersomes against MDR *S. aureus* was investigated. The study’s findings demonstrated that the produced chemical was highly biocompatible, with minimal hemolysis and cytotoxicity. The QRT filled vesicles were reduced in size, measuring roughly 225.5 ± 16.31 nm, with 88% ± 1.52% encapsulation of QRT and a regulated and sustained drug release profile. After being encapsulated in BM nano-vesicles, the MIC value of QRT was reduced to 136 mg/mL. Previously, the MIC value of QRT against *S. aureus* was shown to be 400 mg/mL [138]. The antibacterial activity of QRT-loaded BM nano-vesicles was further validated by AFM pictures, which revealed full deformation of bacteria cell surface shape. The findings of this study imply that the synthesized resorcinarene macrocycle has the potential to improve the treatment potential of QRT [136].

### 2.2. Drug Delivery Platforms

#### 2.2.1. Nanofibers

Nanofibers are fibers having a diameter of around a nanometer. They may be made from numerous of polymers and have a variety of physical characteristics and applications. Nanofiber sizes are determined by the kind of polymer employed and the production procedures [139]. In comparison to their microfiber counterparts, all polymer nanofibers have a large surface area-to-volume ratio, substantial mechanical strength, high porosity and flexibility in functionalization [140]. Some natural polymers including alginate, Cs, silk, keratin, gelatin, cellulose and collagen were explored previously for nanofiber fabrication. Similarly, other Synthetically produced polymers such as polyurethane (PU), PCL, PLA, poly(lactic-co-glycolic acid) (PLGA) and poly(ethylene-*co*-vinylacetate) (PEVA) were all employed in nanofibers [141]. Natural biopolymers have resurfaced as the main bioactive compounds in medical materials applications in recent decades. These polymers often contain biofunctional chemicals that assure biomimetic nature, bioactivity and natural restructuring [142]. They contribute significantly to tissue engineering, particularly in the creation of scaffolds for therapeutic drug delivery. Because of their intrinsic bioactivity, biocompatibility and bioresorbability, novel and natural polymeric materials are aimed at boosting various treatments [143].

Essential oils (EOs) of lemon balm (*Melissa officinalis* L.) and dill (*Anethum graveolens* L.) were recently encapsulated in collagen hydrolysates taken from bovine tendons (HCB) and rabbit skins (HCR), both combined with Cs, for potential wound dressing applications, using the coaxial electrospinning technique. SEM and attenuated total reflectance Fourier transform infrared spectroscopy (ATR-FTIR) were used to analyze the shape and chemical content of electrospun nanofibers [144]. The interaction between components resulted in spherical shapes of collagen hydrolysate-Cs nanofibers with micrometric dimensions. The incorporation of EOs into a collagen-Cs matrix reduced the amount of spherical particles, allowing for increased interactions between components, which is favorable to the administration of bioactive wound dressings [145]. Hua et al. [145] discovered a similar surface shape of Cs-collagen with a spherical structure [145]. Hydrogen bonds were formed via the interaction of amino groups from Cs and carboxyl groups from collagen, according to FTIR analysis. For HCB-Cs compositions, the band intensity for lemon EO and dill EO was about 1640 cm^−1^. The spectra of HCB and HCR containing encapsulated EOs show these bands as well [146]. The interaction between the hydrophobic groups contained in the collagen hydrolysate, Cs and essential oils causes the intensity of certain bands to decrease in the FTIR spectra, indicating a synergic impact between dill EO and lemon balm EO. The bioactive components of dill EO and lemon balm EO are therefore predicted to be present inside the electrospun collagen-Cs complex nanofibers [144].

Disk diffusion assay was used to determine the antimicrobial activity of dill EO and lemon EO, as well as electrospun samples loaded with essential oils against *S. aureus* ATCC 25,923, *E. coli* ATCC 25,922, *E. faecalis* ATCC 29,212, *S. typhimurium* ATCC 14,028, *Candida albicans* ATCC 10,231, *C. glabrata* ATCC 90,028 and *Aspergillus brasiliensis* ATCC 9642. The combination of HCB-Cs with dill EO or lemon balm EO enhanced the antimicrobial activity and the combination with both EOs expanded the antimicrobial activity against *C. glabrata*, *C. albicans*, *E. faecalis* and *S. aureus*, while it diminished the antimicrobial activity against *A. brasiliensis* and *S. typhimurium*. Likewise, HCR-Cs activity was also boosted in the presence of dill EO or lemon balm EO and the inclusion of both oils resulted in more efficient activity against *S. aureus* but less efficient action against *A. brasiliensis* and *E. faecalis* [144]. The antimicrobial activity of collagen nanofibers against *S. aureus*, *E. coli*, *P. aeruginosa* and *C. albicans* [147] or Cs- PVA film against *S. aureus* and *P. aeruginosa* [148] was previously found to be enhanced in the presence of EOs. Another investigation indicated *E. coli* inhibition in the case of Eos loaded in cellulose acetate nanofibers, owing to the fibers’ large exposed surface area as well as the microorganism’s ability to disperse through the network matting, favoring interaction with the bioactive chemicals of EOs [149]. The holes generated in the fibrous network might explain the varied responses of the microorganism to the electrospun fibers. It was possible to permeate the network of electrospun fibers depending on the microbe dimension [149].

In another investigation, electrospinning was used to create composite membranes of PLLA and ZnO NPs comprising 5–40 wt.% of the ZnO NPs (Figure 3). For the first time, fiber-based composite membranes were made using polymer material loaded with up to 40 wt.% ZnO NPs [150]. These membranes were formed through laser ablation in air and having a non-modified surface. A variety of analytical methods, including SEM, XRD, FTIR, UV-vis and photoluminescence spectroscopy, were used to evaluate the membranes’ morphology, phase composition, mechanical, spectral and antibacterial characteristics [150]. As the membranes were doped with greater loads of NM, the composites were discovered to have a number of visible aggregates in the fibers combined with ZnO NPs. Furthermore, because the viscosity of the initial polymer containing NPs decreases, the mean size of the electrospun fibers generated by NPs should decrease when additional NPs are introduced. It must be noticed that there is no proportional correlation between the concentration of additional ZnO NPs and the decrease in fiber widths, which is most likely due to the reality that the conductance of the initial polymer solution varies with its viscosity when more NM is added as filler [151]. *S. aureus* and *E. coli* bacteria were used to test the materials’ antibacterial properties. The results showed that all ZnO-loaded membranes have antibacterial activity, as evidenced by reduction of bacterial growth in both tested bacterium strains. It is worth noting that antibacterial activity against S. aureus is significantly greater than that seen for *E. coli* [150]. This is thought to be due to differences in membrane structure between these two bacteria. The cell membrane of *E. coli* is known to have a more complicated structure than that of *S. aureus*, being thicker and chemically distinct [152]. There are different mechanisms that explain ZnO NPs’ antibacterial effect, including the generation of reactive oxygen species (ROS), the release of Zn^2+^ ions by partial ZnO dissolution and electrostatic adhesion of ZnO NPs to the bacterium membrane. All these methods cause the bacterium cell membrane to degrade, resulting in cell dysfunction and the entry of harmful antibacterial agents such as ROS and Zn^2+^ into the cell [153].

#### 2.2.2. Nanocomposites

Nanocomposites are heterogeneous/hybrid materials made at the nanometric scale by combining polymers with inorganic solids (oxides to clays). Their structures have been discovered to be more complex than microcomposites. Individual property, composition, interfacial interactions, components and structure are significantly influential. Nanocomposites are most commonly made through the in situ growth and polymerization of biopolymer and inorganic matrix. Owing to the rising frequency of drug-resistant bacterial infections, the development of novel antibacterial nanostructures has become paramount [154]. In this regard, Psochia et al. [155] have developed a nanoimprinted PLLA (poly lactic acid) composite films from mesoporous silica NPs, mesoporous cellular foam (MCF). The authors could successfully ameliorate the mechanical and thermal characteristics of PLLA compared to the commercial products by using silica NPs that had already been created as nanocarriers [155]. The films were nanopatterned with thermal nanoimprint lithography (t-NIL) to add antibacterial activity to their surfaces. The t-NIL approach is a promising strategy for creating micro- or nanometer-scaled topographies on materials surfaces such as slopes, walls, columns, and so on, that substantially restrict the accessible area for bacteria to adhere compared to their smooth unpatterned counterparts [156]. The antimicrobial activity of the films was evaluated in vitro against bacterial strains of *E. coli* and *S. aureus* [155]. In comparison to the unpatterned films, the nanostructured surfaces (PLLA/SBA-15 2.5 wt% and PLLA/MCF 2.5 wt%) had reduced cell densities. When compared to its counterparts without topography (PLLA/SBA-15 2.5 wt%, and PLLA/MCF 2.5 wt% TOP), *S. aureus* grew the slowest on surfaces with hierarchical structure (PLLA/SBA-15 2.5 wt% TOP and PLLA/MCF 2.5 wt% TOP). Considering these findings, it can be inferred that the surfaces of nanocomposite PLLA films improved with micro/nano-topographical characteristics hindered considerable bacterial development as compared to those with flat or unpatterned surfaces, since the accessible space for bacteria was reduced [157].

Moreover, Xiao et al. [158] have recently developed functional nanocomposite films based on soy protein isolate that are strengthened with cellulose nanocrystals (CNC) and zinc oxide nanoparticles (ZnONP). In a pork sample, the ZnO-containing films prevented the development of foodborne pathogens (*E. coli* and *S. aureus*) and decreased total viable counts as well as total volatile basic nitrogen levels. [158]. ZnO NP was shown to be more efficient against Gram-negative *E. coli* than Gram-positive *S. aureus*, which was attributed to the bacteria’s cell wall structure. Gram-positive bacteria have a strong cell wall made up of multi-layered peptidoglycan that prevents ZnO NP from penetrating the cells. Meanwhile, gram-negative bacteria have a thin peptidoglycan layer that is covered by an outer membrane, which may help nanoparticle attachment and absorption [159]. Several theories have been offered to explain ZnO NP’s antibacterial action [159,160]. First, zinc ions can penetrate the cell wall and kill bacteria by interacting with the cytoplasmic material. Second, ZnO NP’s electrostatic and direct contacts with the bacterial surface cause structural alterations or bacterial inactivation. Third, the reactive oxygen species produced by ZnO have the potential to harm the bacterial cell membrane. Antibacterial activity of ZnO-containing films was shown against both gram-negative *E. coli* and gram-positive *S. aureus*, and Fresh pork’s shelf life was significantly prolonged [159]. Furthermore, The CNC@ZnO NP film inhibited zinc migration from the film to the substrate system of food. This research provides a theoretical foundation for the design and implementation of the project. Multifunctional active packaging materials are being developed. To increase the efficiency of nanofillers, research should be focused on their mechanisms [159].

### 2.3. Recent Polysaccharides Based Nanomaterials

Drug administration is now the most common and appropriate approach for human medication delivery. However, oral and injectable medicine is not always an efficient technique since there are various issues with the oral drug delivery process, such as a lack of drug absorption, a short residence duration and gastrointestinal instability (GIT). Some medication delivery technologies can be employed to circumvent these disadvantages [160]. Polysaccharide-based nanocarriers have been widely employed in neumerous drug delivery route in recent years, attracting considerable attention since they exhibit several benefits [161]. Polysaccharides are carbohydrates made up of monosaccharides linked together by glycosidic linkages. Polysaccharides, as nanocarriers, can circumvent the reticuloendothelial system and with an increased enhanced stability in blood stream [162]. Furthermore, chemical and biological procedures can be used to modify numerous functional groups in the polysaccharide structure. The hydrophilic groups in the polysaccharide structure, such as carboxyl, hydroxyl, and amino groups, can boost medication bioavailability by promoting cell and tissue adhesion. It is worth mentioning that polysaccharide biological activity is tightly connected to their structure. Polysaccharides having a medium molecular weight and a triple helix structure, show stronger antitumor and immunomodulatory action [163]. Furthermore, polysaccharide sugar group composition, physical characteristics, glycosidic bond types and ratios, branched-chain structure and substitute groups, all have a significant influence on their biological activities [164].

Recently, Dong et al. [165] created an injectable antibacterial hydrogel based on hyaluronic acid (HA) and chlorhexidine (CHX) to treat infections in cardiovascular implanted electronic devices (CIEDs). To achieve a balance of stability and moldability, the HA scaffold was pre-crosslinked with 1,4-butanediol diglycidyl ether (BDDE) before being ground to create a HA microgel (CHA). The antibacterial agent CHX was then cross-linked in the CHA microgel via electrostatic interactions between the CHA and CHX to produce hybrid crosslinked hydrogels (CHA/CHX). These hydrogels demonstrated shear-thinning/self-recovery behaviour, allowing for facile injection into the CIED pocket and good matching with the pocket form without the need for additional space, which is an improvement over previously described approaches. In vitro and in vivo antibacterial experiments revealed that the CHA/CHX hydrogels exhibited high biocompatibility as well as extremely potent antibacterial activity against *S. aureus* and *E. coli*. Because of the poor physical connection between the CHA microgel and the loaded CHX, continuous CHX release from the hydrogel was possible during the acute infection period 7–10 days after implantation [166]. The aforesaid findings suggested that the CHA/CHX hydrogels would be a good choice for the treatment of CIED pocket infections.

In similar manner, HA was studied as a coating for the Ti6Al4V surface, which is commonly utilized for permanent implants in the field of prosthesis and dentistry. Poly(N-vinyl pyrrolidone) (PVP), HA and Cs polymeric antibacterial coatings were produced and compared for Ti6Al4V surface covering. These coatings were adhered to Ti6Al4V surfaces after being conjugated with the well-known bio-adhesive capabilities of the catechol (CA) anchor group [167]. The antibacterial characteristics and cytocompatibility of coated Ti6Al4V substrates against Gram-positive and Gram-negative strains were evaluated, showing the efficiency of these polymeric coatings against bacterial infections for future applications in preserving biomedical implants [167]. Other polysaccharides such as thiol, mannose and galactose have been conjugated with chitosan forming antibacterial nanomaterials. For instance, Mannose functionalized Cs nanosystems (M-CNS) were created by reductively aminating Cs with mannose and then ionic gelation. Surface functionalization of Cs with mannose was shown by changes in zeta potential and distinctive peaks in FTIR spectra. Mannosylated CNS was shown to be somewhat larger (180.5 nm) than CNS (162.7 nm) in zeta-sizer investigations. For M-CNS, the zeta-potential was lowered from +32.2 mV to +25.4 mV [168]. Time-kill, polystyrene adherence, and antibiofilm tests were used to evaluate the antimicrobial efficacy of developed nanosystems as an alternative antibacterial agent against Gram-positive and Gram-negative bacterial infections. Mannose functionalized CNS reduced the development of resistant *E. coli* and *Listeria monocytogenes* while also exhibiting anti-adherence and biofilm breakup action. Furthermore, M-CNS was vulnerable to highly resistant *P. aeruginosa* and *S. aureus*. These findings revealed M-CNS ability against pathogenic, biofilm-forming, multidrug-resistant bacteria, making them an optimum choice for creating alternative medications to treat emerging resistant illnesses [168].

### 2.4. Green Synthesis of Polymeric Nanomaterials

Green nanotechnology combines the concepts of green chemistry and green engineering to create harmless and environmentally friendly nano-assemblies to tackle problems impacting human health or the environment. As a result of the combination of green nanotechnology with medication delivery, a new field of “green nanomedicine” has emerged [169]. The growing interest in green nanotechnology-driven drug delivery systems has enabled the development of various delivery devices such as quantum dots, mesoporous silica nanoparticles, nanostructured lipid carriers and solid lipid nanoparticles [169].

#### 2.4.1. Quantum Dots (QDs)

QDs are nanoscale semiconductor particles with optical and electronic properties that diverge from larger particles own to quantum mechanics [170]. QDs have shown to be a vital tool in the fields of sensing, detection and biological imaging, prompting scientists to create entities for translational and scientific medicine. QDs are extensively used in quantitative detection and imaging because they are powerful fluorescent probes [171]. QDs with therapeutic properties and assimilated targeting have shown to be excellent materials for studying drug delivery in cells and small animals. Because of their outstanding biocompatibility and great optical characteristics, a wide variety of QDs, such as graphene QDs (G-QDs), carbon dots (C-QDs), nitrogen-doped QDs (N-QDs), silicon QDs (Si-QDs), and others, have been created to date. Several medications have been successfully included into QDs in order to deliver them to the precise target spot. Furthermore, typical QD synthesis necessitates the use of heavy metal precursors and organic solvents, both of which are potentially hazardous to the environment. Recently, using dimethyl diallyl ammonium chloride and glucose as reaction precursors, quaternized carbon quantum dots (qCQDs) with broad-spectrum antibacterial activity were manufactured in a simple green “one-pot” process [172]. The antibacterial efficacy of the qCQDs against Gram-positive and Gram-negative bacteria was excellent. qCQDs clearly restored the weight of rats in wounds infected with mixed bacteria, considerably decreased the death of rats from severe infection, and improved the recovery and healing of infected wounds in rat models of wounds infected with mixed bacteria. During the testing stage, biosafety studies verified that qCQDs had no clear harmful or adverse effects. The quantitative proteomics study demonstrated that qCQDs primarily worked on ribosomal proteins in *S. aureus* and dramatically down-regulated citrate cycle proteins in *E. coli* [172].

#### 2.4.2. Mesoporous Silica Nanoparticles (MSNs)

MSNs are silica nanoparticles with holes ranging in size from 2 to 50 nm and an overall diameter of less than 1 µm. Their pore size distribution is compatible with the IUPAC definition of mesoporous, making them attractive materials for applications ranging from environmental chemical removal to biomedicine. MSNs are commonly proposed as drug delivery matrices due to morphological characteristics such as large surface area, pore diameter and easily modifiable surface features. SBA-15, MCM-41 and MCM-48 are the most prevalent mesoporous silica materials with pore sizes of 2–10 nm and two/three-dimensional cubic characteristics [173]. Son and Lee [174] investigated not only the removal rate of cetyltrimethylammonium ammonium bromide (CTAB) from MSNs, but also functional decorations of certain chemical moieties such as polydopamine (PDA) and graphene oxide (GO). The *E. coli* death rate was greatly influenced by the remaining CTAB that was not entirely eliminated from the MSNs during the acid etching procedure. However, for the removal of CTAB from MSNs, the shorter etching duration and higher H_2_O concentration in the etchant boosted *E. coli* survivability, most likely due to a decrease in bioactive H^+^ ions that restrict bacterial growth. Notably, PDA-coated and GO-wrapped MSNs were more hazardous than pure MSNs due to the significant interaction of antibacterial activity of functional decoration (PDA and GO). This research serves as a valuable guideline for the development of biocompatible MSNs for biological applications [174].

#### 2.4.3. Nanostructured Lipid Carriers (NLCs)

NLCs are new pharmaceutical formulas made up of physiologically and biocompatible lipids, co-surfactants and surfactants. NLC has developed as a second-generation lipid nanocarrier as an alternative to first generation nanoparticles throughout time [175]. NLCs were developed to address the issues associated with first-generation SLNs by enhancing loading capacity and preventing drug expulsion. Lipid like materials such as stearic acid, glyceryl palmitostearate, glyceryl dibehenate, caprylic/capric triglycerides, tripalmitin, vitamin E, lauryl polyoxylglycerides, soya lecithin, polyoxylcastor oil and polysorbates are often utilized in the manufacturing of NLCs [176]. The NLCs can be utilized in particulate form by the intestine and transferred to several lymphatic systems organs. The medication can be encapsulated inside the NLCs and delivered via several routes, including intravenous, oral, ocular and pulmonary [177]. In a new study, Gels made from *Mentha pulegium* essential oil (MPO) injected into NLCs (MPO-NLCs) were proposed to speed up the healing of infected wounds. This study looks at MPO-NLCs’ in-vitro antibacterial activity as well as their in-vivo wound healing activity in a BALB/c mouse model. To analyze the influence of MPO-NLCs on tissue bacterial count, wound contraction and healing, histological and molecular parameters were measured at 3, 7 and 14 days after the wound creation [178]. MPO-NLCs had the highest antibacterial activity against *S. epidermidis*, *S. aureus*, *L. monocytogenes*, *E. coli* and *P. aeruginosa*. The application of MPO-NLCs to the skin reduced the inflammatory phase and enhanced the proliferative phase. Furthermore, as compared to the control group, injection of MPO-NLCs boosted the expression of inflammatory mediators such as IL-10, TGF-β and b-FGF while decreasing the levels of NF-κB. MPO-NLCs aided infected wound healing by speeding up the proliferation phase of the wound healing process, lowering the inflammatory phase and boosting antibacterial capabilities. MPO-NLCs have the potential to be exploited as an agent in the treatment of infected wounds [178].

#### 2.4.4. Solid Lipid Nanoparticles (SLNs)

SLN were described as small and spherical particles formed of solid lipids at room temperature, which may be regarded of as ideal crystal lipid matrices capable of containing a medicine or other molecules within their fatty acid chains [179]. SLNs are at the leading edge of nanocarrier technology, with several potential applications in research and clinical medicine and drug delivery. The capacity of SLNs to encapsulate pharmaceuticals into nanocarriers with unique size dependent features provides a novel prototype that may be employed for secondary and tertiary medication targeting [180]. Using RAW 264.7 cells, a recent study developed and tested four fatty acid-SLNs for intracellular transport, accumulation, and disparity in antibacterial activity of their loaded enrofloxacin. The effectiveness of enrofloxacin transport into macrophages by octadecanoic acid SLNs (OAS), tetradecanoic acid SLNs (TAS), hexadecanoic acid SLNs (HAS) and docosanoic acid SLNs (DAS) was 26.1–29.0, 4.7–5.3, 4.5–5.0 and 9.3–10.3 times, respectively in comparison with free drug during 0.25–4 h co-incubation [181]. The longer the fatty acid produced nanoparticles loaded with enrofloxacin were removed, the slower it was cleared and the longer it accumulated in the cells. Confocal microscopy also revealed that as the carbon chain of fatty acids lengthened, more fatty acid SLNs entered the cells with greater accumulation performance and less SLNs absorbed on the cytomembrane. When compared to free enrofloxacin, the bactericidal efficacy of the four fatty acid SLNs against intracellular Salmonella CVCC541 was dramatically increased. These findings suggest that fatty acid SLNs, particularly docosanoic acid nanoparticles, might be excellent nanocarriers for delivering enrofloxacin or other lipid-soluble medicines into cells for the treatment of intracellular bacterial infections [182].

## 3. Polymeric Nanomaterials Combating Bacterial Biofilms

Bacteria that develop in biofilms contribute significantly to the formation of drug resistance [183]. A biofilm is an assembly of bacteria and self-produced extracellular polymeric substances, such as polysaccharides, extracellular DNA and proteins. There are several considerations contributing to improved drug-resistance in biofilms. First and foremost, aminoglycosides and polymyxins are examples of conventional antibiotics have a positive net charge and can interact with negatively charged extracellular polymeric molecules in biofilms, posing a significant barrier antibiotics’ penetration into biofilms. Second, there are bacteria immersed in biofilms have a slow-growing phenotype, characterized by decreased nutrient absorption as well as other harmful chemicals. Third, there are a plethora of enzymes contained in biofilms, which might secreted by the microorganisms [184].

Drug delivery systems (DDS) have advanced significantly in recent decades for eliminating bacteria entrenched in biofilms while avoiding antimicrobial adverse effects on human tissues. They have the ability to address the issues of intrinsic antimicrobial resistance and low antimicrobial penetration into the biofilm. Nevertheless, as a disadvantage, certain DDS frequently display a rapid drug release followed by a delayed drug release after delivered [185]. This type of medication delivery method frequently encounters undesired antimicrobial leakage before reaching infected locations, which might promote antimicrobial resistance. An ideal drug delivery system for biofilm treatment would keep antimicrobial concentrations in blood or tissue around biofilms between the minimum effective concentration (MEC) and the minimum toxic concentration (MTC) for a set period of time, reducing drug toxicity and increasing patient compliance (Figure 4) [186]. However, a variety of parameters, including carrier matrixes, drug-carrier interactions, and drug physicochemical features, may influence DDS release rates [187]. Meanwhile, for future antimicrobial applications, other needs, such as smart, self-regulated release and targeted delivery, are crucial.

A novel antimicrobial surface coating comprised of the synbiotics (prebiotics and probiotics mixture) with potent antibacterial and antibiofilm prospects was recently generated by Harandi et al. [188]. Two *Lactobacillus plantarum* and *Lactobacillus acidophilus*-green Fe_2_O_3_ NPs (Fe_2_O_3_NPs-LAB) hybrids were created for covering the surface of polyvinyl alcohol (PVA)-prebiotic gum arabic-polycaprolacton nanofibers (PVA-GA-PCL) as an emerging class of wound dressing materials. Because of its large molecular weight, mechanism of swelling, decreased solubility and appropriate solvent selection, the electrospinning of GA presents certain obstacles. GA must be mixed with synthetic polymers to circumvent the constraints [189]. Owing to the merits of PVA such as solubility in water, thermal characteristics, low cost, permeability and biocompatibility, It was mixed with GA. The linkages of hydrogen bonding between the hydroxyl and amide groups of polymers (GA and PVA) and the carboxyl groups of Fe_2_O_3_ NPs-LAB might be used to characterize the wrapping mechanism of nanofibers with Fe_2_O_3_ NPs-LAB.

In terms of antibacterial performance, the MICs for *E. coli*, *S. aureus*, *P. aeruginosa* and *C. albicans* ranged from 18.75 to 300 µg/mL, while MBC and MFC ranged from 75 to 300 µg/mL. *C. albicans* had the highest MIC value of 18.75 µg/mL. *S. aureus* and *E. coli* had the greatest growth inhibition rates of 66% and 88%, respectively. In general, the pathogenic strains were sensitive to Fe_2_O_3_ NPs-LAB throughout a wide range. Due to bacterial cell wall differences, *P. aeruginosa* with MIC values of 37.5 and 75 µg/mL was selected as the most sensitive bacteria in contrast to other bacteria. The biofilm of *P. aeruginosa* was the most susceptible (43.73%), whereas *E. coli* biofilm was the most resistant (19.71%). Antimicrobial agents (MNPs and LAB) included in bionanocomposite have a considerable influence on biofilm suppression [190].

The antibiofilm efficiency of MNPs is strongly dependent on their physicochemical qualities and biological activity (for example, size, concentration, antibacterial performance, biosorption, and affinity between the substances and the biofilm). Because of their physicochemical properties, MNPs can adsorb to the biofilm surface, penetrate, and travel inside it, resulting in biofilm inhibition [191]. Furthermore, the kind of pathogenic bacteria impacts biofilm formation. Fungal hyphae, for example, show a high capability for surface attachment as compared to blastospore forms [192]. By secreting antagonistic materials, generating signal molecules in quorum sensing and establishing adverse environmental circumstances for pathogenic bacteria, LAB limit biofilm growth [190]. Furthermore, some probiotic lactobacilli can inhibit the production of *C. albicans* biofilms.

Likely, antibiofilm activity of iron oxide (IO) NPs-Cs nanocomposite (25, 50, 75, and 100 g/mL) against *E. coli* and *S. aureus* was investigated. The results showed that harmful bacteria in biofilms were sensitive to nanocomposite. Furthermore, it was shown that the antibiofilm action of IO NPs is concentration-dependent [193]. In addition, a bioactive covering was created based on patchouli oil and Fe_3_O_4_ NPs to decrease cell number of *S. aureus* biofilm. Ambrogi et al. [194] investigated the antibacterial and antibiofilm capabilities of alginate films using pyrogenic silica-supported silver NPs as prospective wound treatments. First, silica-supported silver NPs (CAB-O-SIL-Ag) were synthesized using a solid-state sintering method that did not need the use of a solvent or reducing agent. Treatment with a crosslinking agent, such as calcium ion, which may operate as an ionic crosslinker, results in films with high mechanical characteristics and resistance, as well as the ability to adsorb water and sustain hydration and wound protection [194]. Calcium ions interact with the alginate’s guluronate blocks, causing chain-chain interactions and the creation of junction zones known as egg box junctions. An external gelation approach was used to create cross-linked films, in which cross-linking happened initially at the film surface, upon contact with Ca solutions and contact duration and calcium concentration are variables that affect gelation degree [195].

Using core shell systems is one approach to immobilize silver NPs. AgNPs are encased within the silica shell’s interior, avoiding aggregation; nevertheless, the danger of this strategy is a reduction in the functionality of the packed materials pyrogenic silica. It has been employed as a substrate for the formation of AgNPs due to its large surface area and the existence of interparticular pores [194]. This matrix was selected for its high specific surface area of roughly 300 m^2^/g, which allows silver to be distributed over a vast surface. Furthermore, the presence of free surface silanols might interact with silver ions, reducing their mobility and influencing NPs development and, as a result, silver release [194]. Silver NPs exhibited an initial rapid release, allowing the release of 2% of the total silver quantity present in the film after 5 h, succeeded by a very gradual release of silver ions. The oxidation of AgNPs closer to the film surface might explain the initial faster release of silver ions, which occurs rapidly when the film is exposed to air. This delayed release might allow for a longer activity while also limiting excessive silver ion concentrations, which could be hazardous to the host [196].

The observation of antibacterial activity in the film contact area, together with the minimal amount of silver liberated, show that antibacterial activity is also exerted by direct contact action of silver NPs embedded in films, as previously hypothesized [197]. Indeed, it has been postulated that nanostructure features, other than the release of Ag^+^ ions, may give rise to intrinsic antibacterial activity, such as ROS production, adsorption of NMs to bacterial cells and changes in bacterial membrane permeability [198]. Many bacteria are located on the skin’s surface in the sessile form as biofilms. A substantial body of data implies that they are involved in chronic inflammation, which causes wound healing to be delayed. As a result, the antibiofilm activities of films were also investigated, which demonstrated a mass biofilm decrease of around 30.7% for *S. epidermidis* and 57.2% for *P. aeruginosa*. Because of disaggregation throughout the test, it was not able to evaluate film 12’s antibiofilm activity [194].

## 4. Natural vs. Synthetic Polymers

Natural polymers such as proteins and nucleic acids are found abundantly in nature and can be naturally or chemically extracted from their sources [199]. Proteins, chitin, pectin and cellulose are some examples of naturally occurring polymers [200]. On the other hand, synthetic polymers are man-made polymers obtained mostly from petroleum oil. Natural biopolymers have resurfaced in recent decades as main bioactive compounds utilized in medicinal material applications [142]. Albumin, chitosan, alginate, collagen and gelatin are the most often employed natural polymers in DDS. The most significant characteristics of natural polymers are 3D geometry, antigenicity, biocompatibility, bioactivity, intrinsic structural similarity and non-toxic byproducts of biodegradation [201]. To create natural polymer-based nanodevices, many processes are used [199]. The nano-fabrication approach is chosen depending on the nature of the active ingredient, the polymer and the NPs characteristics such as EE, zeta potential and particles size Among the most common approaches are ionotropic gelation, emulsion-cross linking and coacervation [199]. According to the database survey, the use of polymers (synthetic/natural) is strongly suggested since the characteristics of NMs may be adjusted by utilizing different polymers and co-polymers. However, microbial contamination, limited tunability, immunogenic reactivity, unpredictable rate of degradation, and inadequate mechanical strength limit their applicability of synthetic polymers for hard tissue regeneration.

Natural polymers have significant benefits over synthetic materials, such as exceptional biocompatibility, effective biological function, lower toxicity, enhanced cell response when connected with cells, improved bioactivity and extreme hydrophilicity. On the other hand, they are frequently limited in their applicability due to their poor engineering qualities such as mechanical and thermal properties [3]. Contradictorily, synthetic polymers have many pluses, including better control over chemical composition, significantly in relation to processability and mechanical characteristics, but scaffold products lack bioactivity, have low cell attachment capacity, are hydrophobic, and have reduced surface cell recognition [3].

It is argued that no one material meets all of the requirements for being acceptable for several applications. Instead, a composite including both natural and synthetic polymers can offer a material that meets all criteria, including accepted thermal and mechanical properties, biocompatibility and biodegradability [202]. Several researchers have examined the use of a specific blend of natural and synthetic materials for the fabrication of tissue scaffolds in order to take use of natural materials’ inherent biocompatibility and synthetic polymers’ physicochemical characteristics. Combining natural and synthetic polymers is a versatile strategy for designing more effective platforms with enhanced physical and biological properties [143,203]. They have been mixed to take use of their advantageous features in order to offset the shortcomings of each type of material [204].

For instance, Zhang et al. [205] fabricated a gelatin/PCL/Cs sandwich-like fiber mat through electrospinning. These composites exhibited desirable physicochemical features, such as adequate porosity (50%), pore size (10 μm), acceptable mechanical stability as well as satisfactory swelling and hydrophilicity. Degradability studies in vitro and in vivo confirmed appropriate degradation rates, which more closely approximated the tissue regeneration process. The scaffolds had acceptable biocompatibility, did not impair cell attachment ability and caused high amounts of collagen production in the cell viability experiment and Sirius red staining [205]. Similarly, an electrospinning method was used to create PLA/cellulose nanocrystal (CNC) composite scaffolds to test the effect of CNCs on the biocompatibility and osteogenic potential of PLA [203]. When compared to pure polymer, the mechanical characteristics of the composite were significantly improved and the thermal stability of composite scaffolds was greater. This is because the polymer chains and CNCs have stronger interactions. Notably, the produced composite resulted in outstanding adhesion and proliferation, demonstrating their better biocompatibility [203]. An electrospinning approach was used to create PLA/CNC composite scaffolds to test the effect of CNCs on the biocompatibility and osteogenic potential of PLA. When compared to pure polymer, the mechanical characteristics of composite scaffolds were significantly improved. The authors postulated that the significant amendment in the mechanical strength is owing to the stronger interactions between the polymer chains and CNCs [206]. In addition, the composite scaffolds outperformed pure polymer in terms of heat stability. Surprisingly, the presence of the manufactured composite scaffolds resulted in outstanding adherence and proliferation, showing their improved biocompatibility [203]. Taking all criteria in to consideration, new trends towards biobased polymers via post-synthetic modification is paramount.

## 5. Post-Synthetic Modification of Natural Polymers

Biopolymers are polymers derived from natural sources that are either chemically synthesized from a biological substance or completely synthesized by living cells [207]. Because of their low heat resistance, most biopolymers are thermoplastic with restricted processability. These traits might lead to very poor mechanical properties with little extension at break values, limiting their application range [208]. Several in vivo techniques for biopolymer modifications are exist, ranging from polymer synthesis to material performance enhancement after synthesis. The approaches for enzymatic and chemical polymer modification attempt to change the structures of biopolymers and hence their properties while keeping biodegradability and biocompatibility [209].

### 5.1. Natural Polyesters

Naturally generated PHAs biopolymers have limited medicinal uses, because of their brittle and hydrophobic characteristics [7]. P(3HB-*co*-3HV) copolymer was synthesized utilizing modified *E. coli* YJ101 and then was functionalized with ascorbic acid by *C. antarctica* lipase B-mediated esterification (Figure 5Ai) [210]. When compared to the standard P(3HB-*co*-3HV), the copolymer P(3HB-*co*-3HV)-ascorbic acid had a lower degree of crystallinity (9.96%), a higher hydrophilicity (68°) and a greater thermal breakdown temperature (294.97 °C) [210]. Furthermore, P(3HB-*co*-3HV)-ascorbic acid biomaterial demonstrated a 14% 1,1-diphenyl-2-picryl-hydrazyl (DPPH) scavenging efficacy and a 1.6 times improvement in biodegradability when compared to P(3HB-*co*-3HV). The addition of functional groups to PHA polymer characteristics may be an useful way to boost their biodegradability, economic worth, and essential uses in the medical area [210].

In the same context, the reaction of methyl salicylate with methacrylic anhydride produced a new biobased vinyl monomer. The resulting salicylate-based monomer was polymerized utilizing two or three trithiocarbonate functionalized poly(3-hydroxybutyrate) (PHB) macro agents in reversible addition fragmentation transfer (RAFT) polymerizations (Figure 5Aii) [211]. According to DSC measurements, the resultant block copolymers had a glass transition temperature (*Tg*) of 5 °C and a melting temperature (*Tm*) of 160 °C. TGA findings showed that the produced block copolymers were more thermally stable than PHB. The overall rate constants of RAFT polymerization utilizing tri-functionalized PHB macro agent were 2.58 × 10^−4^, 1.17 × 10^−4^ and 0.50 × 10^−4^ L mol^−1^s^−1^ for 70, 80 and 90 °C, respectively [211]. Because PHB is a naturally degradable polyester, the comparatively low molar weights of the resulting block copolymer (4600 to 8200 Da) might be attributed to the breakdown of PHB macro RAFT agents during synthesis. The molar mass of methacrylated methyl salicylate (KzSA) blocks increased as polymerization time increased. These findings agreed with the RAFT polymerization kinetics. Additionally, It was noted that the activation energy was 18.6 kcal K^−1^mol^−1^. The total rate constant of the resulting monomer’s homopolymerization was determined to be 1.14 × 10^−4^ Lmol^−1^s^−1^. The intriguing popcorn structure of the fracture surface of the generated PHB-KzSA block copolymers might be relevant for medical applications. Furthermore, the produced block copolymers retained the characteristics of both PHB and salicylates, which might allow for future molecular design toward diverse biomaterials. Furthermore, the improved thermal stability of the PHB-PKzSA block copolymers may ease the processing technique.

Relevant biological activities of the generated block copolymers are now being researched in order to further evaluate their application potential [211].

### 5.2. Polyamines

Among polyamines Cs is likely to obtain intriguing characteristics through polymeric modification. Pyridoxal derivatives of Cs with varying degrees of substitution (DS) were synthesized from low-, moderate-, and high-molecular-weight Cs using pyridoxal reaction followed by NaBH4 treatment (Figure 5Bi) [212]. The derivative with the highest DS and a moderate molecular weight displayed the most antibacterial efficacy against *S. aureus* and *E. coli* [213]. This derivative’s NPs generated using ionic gelation are non-toxic and have a strong in vitro antibacterial action that marginally outperforms ampicillin and gentamicin [212]. Furthermore, in order to improve the bioactivity of Cs, Zhang et al. [214] synthesized a novel series of Cs derivatives (Figure 5Bii). To illustrate, Cs was reacted with methylchlorofonmate to produce N-methoxyformylated Cs, which was then converted into N-pyridylurea Cs derivatives via an amine-ester exchange reaction. Furthermore, N-pyridylurea Cs derivatives were synthesized by interacting with iodomethane to produce quaternized N-pyridylurea Cs derivatives [214]. In the meanwhile, the antioxidant activity of Cs derivatives was tested in vitro. The antioxidant activity of quaternized N-pyridylurea Cs derivatives was instantly increased when compared to N-pyridylurea Cs derivatives after quaternization with iodomethane. L929 cells were also used in the CCK-8 experiment to examine the cytotoxicity of Cs and its derivatives where all samples exhibited reduced cytotoxicity values. These findings showed that the new pyridylurea-functionalized Cs derivatives could be an innovative antibacterial/antioxidant attributes for DD [214].

### 5.3. Polysaccharides

Curdlan is one of the most popular polysaccharides which has been under investigation for several decades. This (1→3) glucan polysaccharide has a characteristic thermal gel-forming feature and is edible; it has been certified for use in the food sector by the Food and Drug Administration (FDA). However, curdlan’s low solubility limits its usage, thus many researchers have focused on curdlan modification, resulting in a wide range of curdlan derivatives (Figure 5C) [215]. Curdlan has essential bioproperities such as anti-coagulation, anti-cancer, anti-virus and immune-regulation. Thus, tremendous efforts are needed in the area of post- synthetic modifications of curdlan to enhance its aqueous solubility in water for DD platforms [216]. Curdlan’s low solubility is often attributed to its strong intramolecular or intermolecular hydrogen bonding. The increased water solubility broadens the possibilities for creating curdlan derivatives [217]. Chemical alteration is one method for increasing water solubility. Curdlan’s sugar residue has three hydroxy groups (C2, C4 and C6) that can be exploited as modification sites [217]. Non- selective modification of curdlan include phosphorylation, sulfation and carboxymethylation. These early studies considerably enhanced curdlan solubility, however the majority of the alteration approaches were nonselective [218]. As a result, the structure–function connections were poorly addressed, despite the fact that they are critical for understanding the mechanism of bioactivities and the rheological characteristics of curdlan and its derivatives. As a result, high regioselectivity synthetic techniques are required. Recently, chemical crosslinking was utilized to create ionic hydrogels from curdlan and phosphorylated curdlan [219]. The researchers used 1,4-butanediol diglycidyl ether (BDDE) as a chemical crosslinker to create hydrogels from curdlan and different quantities of phosphorylated curdlan and then investigated the effect of the ionic component on the physicochemical features. At a high pH, the crosslinking process promotes the interaction of the BDDE with polysaccharide hydroxyl groups, since deprotonated hydroxyls are stronger nucleophiles capable of forming persistent ether bonds with cross-linker epoxide groups [220]. The swelling ratio rose from 9 to 16 g/g, tetracycline hydrochloride release increased from 50% to 85% and Young’s modulus lowered from 35 to 14 kPa by increasing the ionic concentration from 1.1 to 3.1 meq/g hydrogel. The diffusion coefficient and drug transport process were studied using mathematical models, with the Ritger–Peppas model providing the best match. The results indicate that these hydrogels might be used in transdermal DD [219].

## 6. Opportunities and Future Prospects

To attain the required performance, a better knowledge of the interactions and mechanisms involved in the construction of polymers hybrid systems is critical, with an emphasis on the following areas: to maximize therapeutic loading while maintaining the integrity of each component, the NM component must be effectively incorporated; mechanical performance must be tunable to match application-specific requirements, particularly for all biomedical applications [221]. In addition, more investigations are needed in the area of eradicating bacterial biofilms in intensive care units and catheters with polymer-based MNPs other than conventional antibiotics with more cytotoxicity studies. In fact, there is a pressing medical demand for novel bioactive agents against hitherto untreatable or difficult to treat diseases such as COVID-19 [222], AIDS, MDR bacterial infections and cancers [223]. Thus, new hybrids or variants generated from post-synthetic chemical modification should undergo toxicity profiles, biodegradability studies and drug interactions prior in vivo studies for antibacterial, anti-cancer, immunomodulatory applications in DD. We strongly believe that in the future more super structures stemmed from polymer modification will serve as DD powerful tools.

## 7. Conclusions

We have discussed numerous emerging classes of NM that have the potential to be used for a wide range of applications such as food safety, pharmaceutics and biomedicine. The structures described in this review have shown the ability to excel at modulating drug release kinetics, assisting localized drug delivery, combating bacterial resistance and biofilm development as well as functioning as tissue scaffolds or food packages. Biomaterials in general have become indispensable tools for nanodelivery. MNPs could be generated safely by microorganisms such as Lactobacillus sp., displaying an impressive gelation capability with gum Arabic and PVA for transdermal delivery. Nanoimprinting approach has proven to be superior in providing the nanosystem with new feature such as antibacterial capacity. Adding to this, weakly charged antibiotics could be targeted with considerable payloads via using micellar nanostructures. Currently, many biodegradable polymers of natural and synthetic origin have been established for use as biomaterials and selecting a polymer for a certain purpose requires careful study of the cellular environment and interactions. However, synthetic polymers possess concerning intrinsic demerits in terms of their toxicity, non-biodegradability and high cost. Thus, there is no surprise that the current interest in neat polymer modifications by chemical and enzymatic techniques emerge from the need for biocompatible and biodegradable polymers having unique or tailored physical and thermal traits. Depending on the kind of polymer to be modified and its intended applications, optimal modification necessitates intelligent adjustment of process parameters such as reactant or sample concentration, treatment dose, catalyst loading, exposure period, and others. Overall, adding one or more active groups could be the key for unlocking polymer/polymer/drug interactions mysteries.

## Figures and Tables

**Figure 1 ijms-23-04101-f001:**
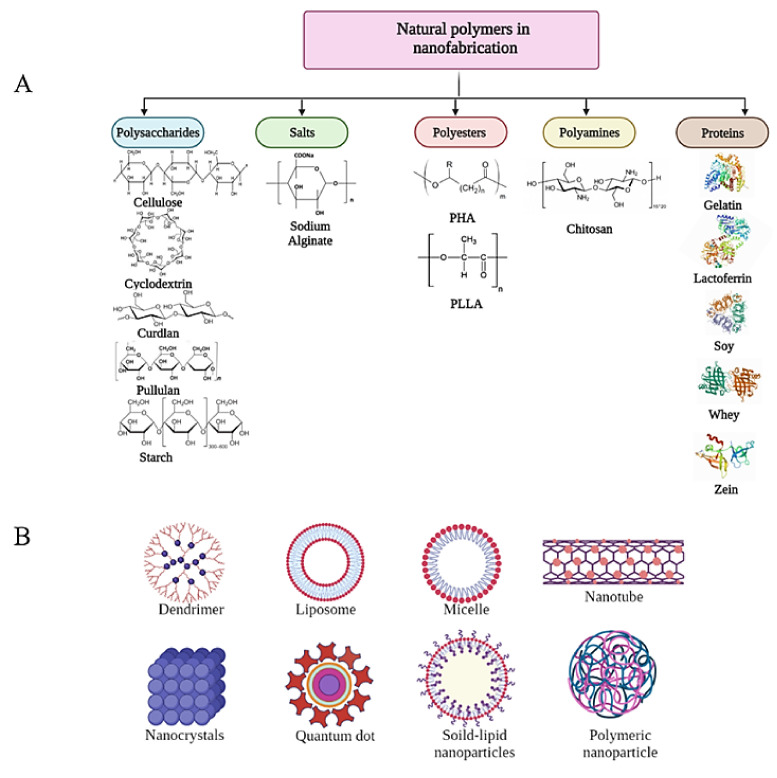
Classification of natural polymers utilized in NMs fabrication (**A**), and their potential structures for bioactive compounds encapsulation (**B**).

**Figure 2 ijms-23-04101-f002:**
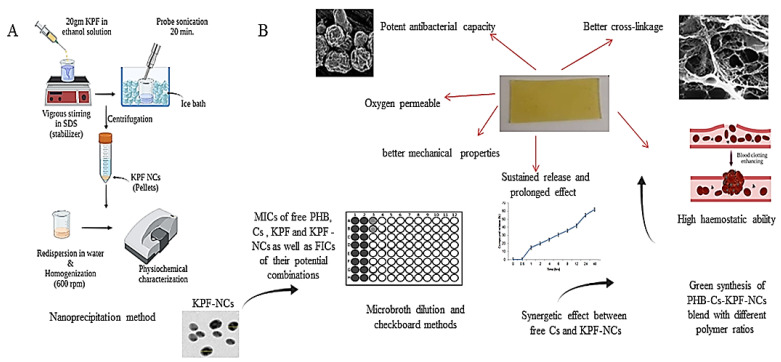
Nanoprecipitation method for KPF-NCs preparation (**A**) and PHB-Cs-KPF-NCs blend with different capacities (**B**).

**Figure 3 ijms-23-04101-f003:**
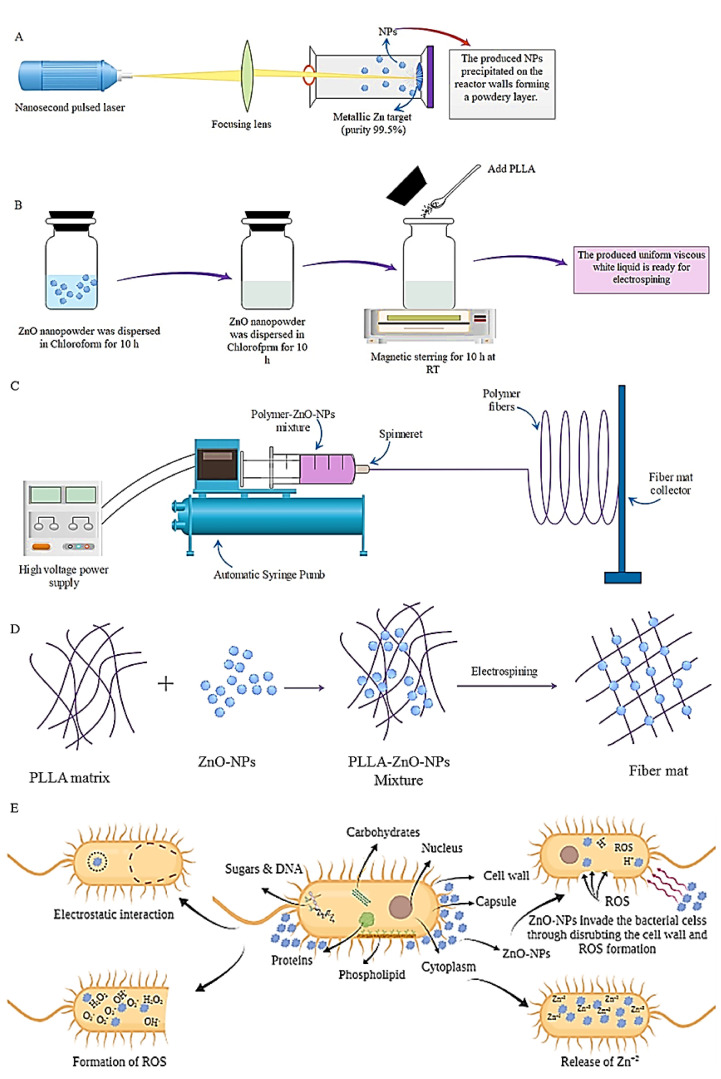
Schematic illustration for the preparation of ZnO-NPs via the pulsed laser ablation in air technique (**A**), fabrication of polymeric -NPs complex (**B**), formation of polymer loaded fibrous mat (**C**), schematic illustration of the ZnO-NPs distribution within fiber mat (**D**) and potential antibacterial mechanisms of ZnO-NPs (**E**).

**Figure 4 ijms-23-04101-f004:**
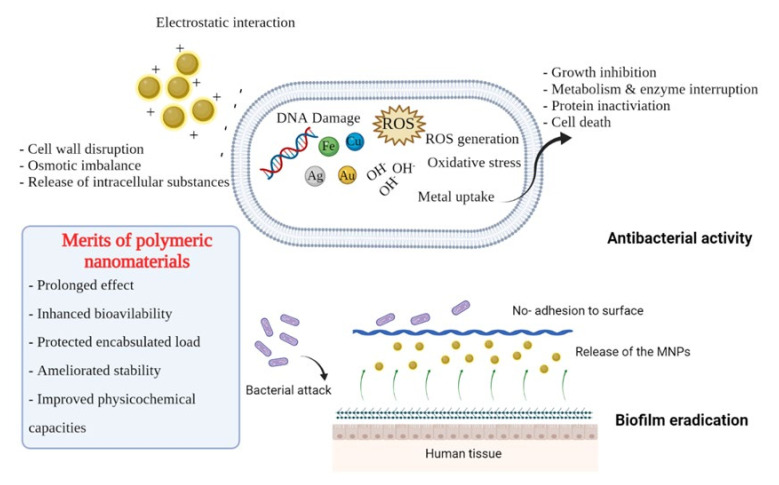
Schematic diagram of antibacterial and antibiofilm mechanisms of biobased polymeric nanomaterials.

**Figure 5 ijms-23-04101-f005:**
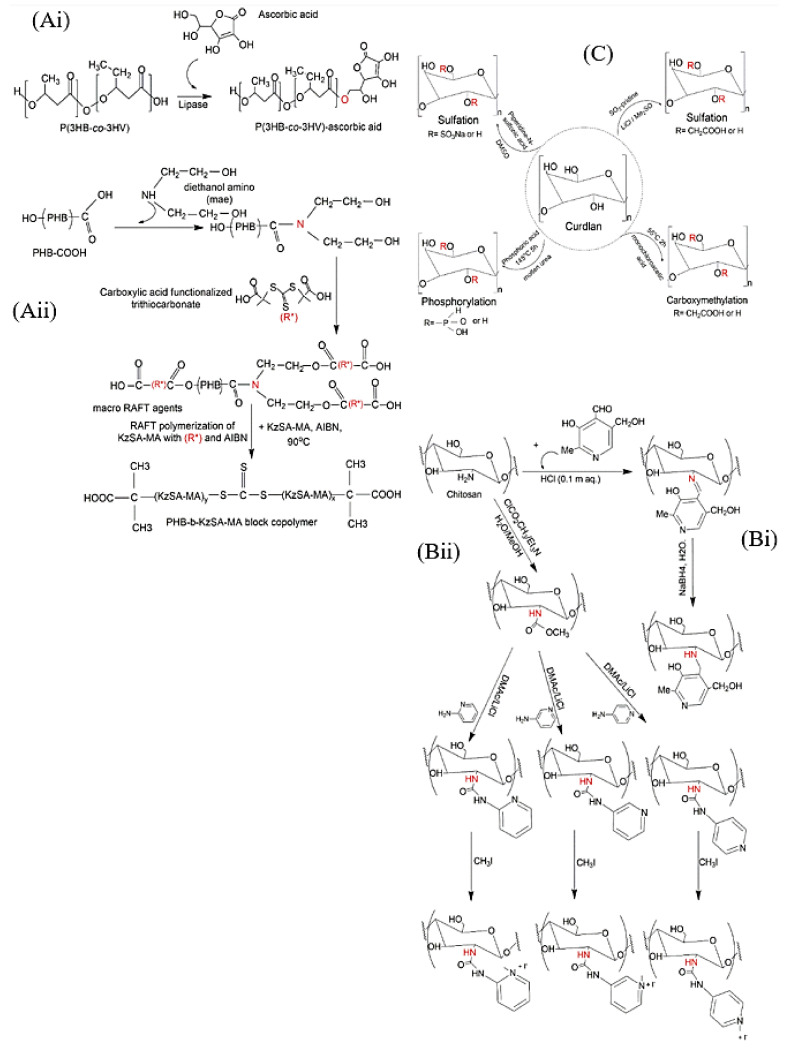
Chemical illustration of post-synthetic modification of three natural polymers, Enzyme mediated modification of P(3HB-co-3HV) (**Ai**), Biobased block copolymer synthesis using a novel methacrylated methyl salicylate and PHB (**Aii**), Cs pyridoxal derivatives are synthesized (**Bi**), Synthetic strategy of the novel pyridylurea-functionalized Cs derivatives (**Bii**) and Curdlan non-selective modification techniques (**C**).

## Data Availability

Not applicable.

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
