# Peer review of "Naturally-Sourced Antibacterial Polymeric Nanomaterials with Special Reference to Modified Polymer Variants"

_ijms, 2022, doi:10.3390/ijms23084101_

Round 1

Reviewer 1 Report

Minor revision

I suggest the author discuss  and add the schematic diagram of the antimicrobial and antibiofilm mechanism of biopolymers  and biopolymeric nanomatrials  in  this review paper.

Author Response

Reviewer 1

Comment:

I suggest the author discuss  and add the schematic diagram of the antimicrobial and antibiofilm mechanism of biopolymers  and biopolymeric nanomatrials  in  this review paper.

Response:

We appreciate the reviewer’s suggestion. A new schematic diagram has been designed and added. (Please refer to Figure 4, Page 24) 

Reviewer 2 Report

The manuscript titled, Naturally-sourced Antibacterial Polymeric Nanomaterials with Special Reference to Modified Polymer Variants, is of good interest to wide range of readers. However, some changes ae advised and need to be addressed to improve the quality of content,

  1. The grammar and sentence structure needs thorough revision. The sentence structures do seem complex in some parts but lack the grammatical orientation.
  2. In section 2, a separate heading should be added to give an overview of the hyaluronic acid based nanomaterials. It possesses antibacterial properties and have been investigated in the antibacterial coatings as well. For Example, Del Olmo, Jon Andrade, et al. "Antibacterial catechol-based hyaluronic acid, chitosan and poly (N-vinyl pyrrolidone) coatings onto Ti6Al4V surfaces for application as biomedical implant." International Journal of Biological Macromolecules 183 (2021): 1222-1235  ,  Dong, Quanbin, et al. "Hyaluronic acid-based antibacterial hydrogels constructed by a hybrid crosslinking strategy for pacemaker pocket infection prevention." Carbohydrate Polymers 245 (2020): 116525.      
  3. Similarly, thiol, mannose, galactose conjugated chitosan nanomaterials as anti-bacterial can also be included in the discussion.
  4. Under some headings, green technology has been mixed with the concept of polymeric nanomaterials. It will be a good approach to include the examples of green synthesis under a separate sub heading for better understanding.                                                     

Author Response

Reviewer 2

The manuscript titled, Naturally-sourced Antibacterial Polymeric Nanomaterials with Special Reference to Modified Polymer Variants, is of good interest to wide range of readers. However, some changes ae advised and need to be addressed to improve the quality of content,

The reviewer’s comment is highly appreciated. The required changes have been addressed to improve the quality of the manuscript. 

Comment:

The grammar and sentence structure needs thorough revision. The sentence structures do seem complex in some parts but lack the grammatical orientation.

Response:

The grammar of the entire manuscript has been check and revised by an English native speaker.

Comment:

In section 2, a separate heading should be added to give an overview of the hyaluronic acid based nanomaterials. It possesses antibacterial properties and have been investigated in the antibacterial coatings as well. For Example, Del Olmo, Jon Andrade, et al. "Antibacterial catechol-based hyaluronic acid, chitosan and poly (N-vinyl pyrrolidone) coatings onto Ti6Al4V surfaces for application as biomedical implant." International Journal of Biological Macromolecules 183 (2021): 1222-1235  ,  Dong, Quanbin, et al. "Hyaluronic acid-based antibacterial hydrogels constructed by a hybrid crosslinking strategy for pacemaker pocket infection prevention." Carbohydrate Polymers 245 (2020): 116525.     

Similarly, thiol, mannose, galactose conjugated chitosan nanomaterials as anti-bacterial can also be included in the discussion.

Response:

The reviewer’s comment is highly considered. A separate heading entitled “Recent polysaccharides Based nanomaterials” has been added. (Please refer to Page 20)  

Comment:

Under some headings, green technology has been mixed with the concept of polymeric nanomaterials. It will be a good approach to include the examples of green synthesis under a separate sub heading for better understanding.

Response:

The reviewer’s comment is highly considered. A separate heading entitled “ Green synthesis of polymeric nanomaterials” has been added. (Please refer to Page 21)